# MCTS2 and distinct eIF2D roles in uORF-dependent translation regulation revealed by in vitro re-initiation assays

Romane Meurs [1], Mara De Matos[1], Adrian Bothe [2], Nicolas Guex [3], Tobias Weber [4], Aurelio A Teleman [4], Nenad Ban [2] & David Gatfield [1]✉

## Abstract

Ribosomes scanning from the mRNA 5′ cap to the start codon may initiate at upstream open reading frames (uORFs), decreasing protein biosynthesis. Termination at a uORF can lead to re-initiation, where 40S subunits resume scanning and initiate another translation event downstream. The noncanonical translation factors MCTS1-DENR participate in re-initiation at specific uORFs, but knowledge of other trans-acting factors or uORF features influencing re-initiation is limited. Here, we establish a cell-free re-initiation assay using HeLa lysates to address this question. Comparing in vivo and in vitro re-initiation on uORF-containing reporters, we validate MCTS1-DENR-dependent re-initiation in vitro. Using this system and ribosome profiling in cells, we found that knockdown of the MCTS1-DENR homolog eIF2D causes widespread gene deregulation unrelated to uORF translation, and thus distinct to MCTS1-DENR-dependent re-initiation regulation. Additionally, we identified MCTS2, encoded by an *Mcts1* retrogene, as a DENR partner promoting re-initiation in vitro, providing a plausible explanation for clinical differences associated with *DENR* vs. *MCTS1* mutations in humans.

**Keywords** uORF; Re-Initiation; DENR-MCTS1; eIF2D; In Vitro Translation
**Subject Categories** RNA Biology; Translation & Protein Quality

## Introduction

Upstream open reading frames (uORFs) are abundant, short regulatory elements found within the 5′ UTRs of an estimated 50% of human transcripts (Wethmar, 2014). uORF translation has long been recognised as a mechanism to tune the translation efficiency (TE) of the protein-coding sequence (CDS) under physiological and stress conditions (Clemm von Hohenberg et al, 2022; Hinnebusch et al, 2016; Jousse et al, 2001; Lee et al, 2009), and several cases of

misregulated uORF-dependent translational control have been linked to human diseases including cancer (Hornig et al, 2016; Occhi et al, 2013; Schulz et al, 2018; von Bohlen et al, 2017). Because uORFs intercept scanning ribosomes and reduce the probability that they engage in initiation downstream, they are generally inhibitory to protein biosynthesis. Two main mechanisms are implicated in ensuring that CDS translation can take place nevertheless: first, in a process called leaky scanning, the 43S preinitiation complex can scan across the uORF start codon, thus ignoring the uORF. Second, the ribosome can translate the uORF, terminate at its stop codon, and then engage in the process of re-initiation, where the small ribosomal subunit remains associated with the transcript to resume scanning to a downstream start codon and initiate another round of translation (Wethmar, 2014). In contrast to canonical initiation, for which many implicated initiation factors and mechanistic details are known, our understanding of the events leading up to re-initiation has remained incomplete. Significant knowledge gaps remain concerning the *trans*-acting protein machinery and the extent to which it overlaps with canonical initiation. Additionally, the specific *cis*-acting mRNA and uORF sequence elements that influence the multistep re-initiation process—from partial ribosome recycling at the uORF termination codon, to the restoration of 40S scanning competence, and the eventual assembly of a new initiation event—are not well understood.

A peculiar case of re-initiation-competent uORFs that has been studied contains only a single amino acid—i.e. the start codon—immediately followed by a stop codon (termed '1-aa uORF' or 'start-stop uORF' in the following) (Schleich et al, 2017; Schleich et al, 2014). Briefly, on the initiating ribosome—through the activities of several initiation factors, including eIF2 that recruits Met-tRNA$^{Met}_i$, followed by extensive eIF remodelling, GTP hydrolysis and eventual 60S subunit joining that all occur in a highly coordinated series of events (Lapointe et al, 2022)—the initiator-tRNA is directly positioned in the ribosomal P-site. Hence, on 1-aa uORFs the stop codon is already placed in the A-site, ready for termination and without the ribosome ever entering into elongation. Hence, and as shown for eIF3 (Mohammad et al, 2017), some canonical initiation factors from the uORF initiation event will

[1]Center for Integrative Genomics, University of Lausanne, 1015 Lausanne, Switzerland. [2]Department of Biology, Institute of Molecular Biology and Biophysics, ETH Zurich, 8093 Zurich, Switzerland. [3]Bioinformatics Competence Center, University of Lausanne, 1015 Lausanne, Switzerland. [4]Division of Signal Transduction in Cancer and Metabolism, German Cancer Research Center (DKFZ), Heidelberg, Germany. ✉E-mail: david.gatfield@unil.ch

likely still be available on the ribosome for subsequent re-initiation. In vivo, certain eIFs have been seen to remain associated with elongating ribosomes for roughly 12 codons, potentially enabling re-initiation after short uORFs but less so for long uORFs (Bohlen et al, 2020a).

Heterodimeric MCTS1-DENR is a non-canonical initiation factor conserved from yeast to mammals and has been identified as the first, re-initiation-specific factor with conserved activity in *Drosophila*, mouse and human (Castelo-Szekely et al, 2019; Schleich et al, 2017; Schleich et al, 2014; Skabkin et al, 2010; Skabkin et al, 2013; Zlotorynski, 2014). In vitro, MCTS1-DENR shows the double activity of being able to recruit and release tRNA from the ribosomal P-site (Dmitriev et al, 2010; Skabkin et al, 2010). These findings have led to the idea that MCTS1-DENR may either act to remove the remaining, deacylated tRNA from the P-site post-termination and after subunit splitting, thereby allowing the 40S subunit to regain the scanning competence that is a prerequisite for re-initiation; and/or to mediate the subsequent delivery of a new Met-loaded initiator tRNA molecule to the 40S subunit. The yeast orthologues, Tma20-Tma22, appear to mainly function in post-termination tRNA removal and 40S ribosome recycling without promoting subsequent re-initiation (Young et al, 2018; Young et al, 2021). In animal model systems, DENR loss-of-function followed by ribosome profiling has been used to identify CDS translation efficiency changes indicative of altered re-initiation rates, thus identifying likely direct MCTS1-DENR targets (Bohlen et al, 2020b; Castelo-Szekely et al, 2019; Schleich et al, 2017; Schleich et al, 2014). These studies revealed that only a small fraction of all uORF-containing transcripts undergo MCTS1-DENR-dependent re-initiation and, while start-stop uORFs are highly represented among targets, longer uORFs are associated with MCTS1-DENR-dependent re-initiation as well. As a first hint towards the selectivity of MCTS1-DENR for certain uORFs, an enrichment for specific penultimate uORF codons was observed among targets in HeLa cells (Bohlen et al, 2020b) and in yeast (Young et al, 2021), supporting the hypothesis that MCTS1-DENR removes deacylated P-site tRNA from the terminated ribosome and that the efficiency of this process may be tRNA-specific.

The protein eIF2D is a close homologue of MCTS1-DENR, containing the same domains encoded on a single polypeptide. Like MCTS1-DENR, eIF2D (then still under the name LGTN, for Ligatin) was initially identified as a recycling and initiation factor based on its in vitro activities (Skabkin et al, 2010), and the yeast orthologue, Tma64, has been noted for its role in 40S subunit recycling, too (Young et al, 2018). Due to these similarities, eIF2D is frequently referred to as a re-initiation factor as well, and evidence for comparable functions of MCTS1-DENR and eIF2D on uORFs come from studies on *Atf4* regulation in *Drosophila* and in HeLa cells (Bohlen et al, 2020b; Vasudevan et al, 2020). However, eIF2D does not regulate an MCTS1-DENR-dependent synthetic model start-stop uORF in vivo (Schleich et al, 2017), suggesting that the two factors may be active on distinct sets of uORFs and transcripts. Furthermore, it has recently been reported that the single knockout of yeast Tma64/*Eif2d* does not recapitulate the 40S recycling defect observed in the single knockout of Tma22/*Denr* (Young et al, 2021). Taken together, the involvement of eIF2D in re-initiation and/or recycling still needs to be formally established (Grove et al, 2024).

In this study, we have combined ribosome profiling, in vivo reporter assays in cells, and experiments using a cell-free translation system to investigate MCTS1-DENR- and eIF2D-dependent re-initiation activity. Using the cellular in vivo and in vitro re-initiation assays, we estimated fluxes of ribosomes on the 5′ UTRs of two identified DENR-dependent model transcripts, *Klhdc8a* and *Asb8*, and addressed the effect of penultimate codon identity. Ribosome profiling from DENR and eIF2D loss-of-function cells pointed at different and distinct roles for the two factors in gene expression regulation and, importantly, indicated that eIF2D does not act as a general uORF re-initiation factor. Finally, we identified MCTS2 as an alternative DENR heterodimerisation partner in vivo and we analysed and validated its activity in our in vitro assay.

## Results

### Upstream open reading frame-containing transcripts *Klhdc8a* and *Asb8* are regulated by DENR in vivo and suitable for model reporter design

To gain insights into the mechanism and targets of DENR-dependent re-initiation, we first identified uORF-containing transcripts for which CDS translation was altered after shRNA-mediated knock-down of *Denr* in murine NIH/3T3 cells (Fig. 1A). To this end, we carried out ribosome profiling (Ribo-seq) and RNA-seq from cells treated with *Denr* shRNA that we compared to cells receiving two different control shRNAs, con1 and con2 (Fig. 1A). From the obtained data, we calculated translation efficiencies (TEs, ratio CDS footprints to RNA) transcriptome-wide, thus identifying 223 and 6 transcripts, respectively, with significantly reduced and increased TE (applying FDR-corrected $p < 0.1$) (Fig. 1B; Dataset EV1). The strong bias for reduced TEs was in line with our expectations for direct targets of a factor facilitating re-initiation on the CDS after uORF translation, and with previous findings (Bohlen et al, 2020b; Castelo-Szekely et al, 2019); we further noted that the annotated 5′ UTRs of transcripts with reduced TE were significantly longer than would be expected from the transcriptome-wide distribution of 5′ UTR lengths (Fig. 1C), compatible with increased uORF content. Indeed, annotation of translated uORFs from the footprint data showed strong enrichment among the *Denr*-responsive transcripts (Fig. 1D). The lengths of the identified uORFs showed a broad distribution; 1-aa/start-stop uORFs were most abundant but many longer, elongating uORFs could be found on the *Denr*-responsive transcripts as well (Fig. 1E). We selected two transcripts to design re-initiation reporters, representing an elongating uORF (*Klhdc8a*, *Kelch domain containing 8A*; 13 amino acid uORF) and a start-stop uORF (*Asb8, Ankyrin repeat and SOCS box-containing 8*) (Fig. 1B). Both mRNAs presented significantly reduced TE upon *Denr* knockdown (decrease by ~70% for *Klhdc8a* and by ~30% for *Asb8*; Fig. 1F) and, moreover, their 5′ UTRs contained only one strongly translated, AUG-initiating uORF (Fig. 1G; Appendix Fig. S1A,B) which is conserved in humans and other mammals (Appendix Fig. S1C,D). Visual inspection of footprint distributions on the transcripts, indicating decreased coverage on the CDS and comparable or even increased coverage on the uORF (Fig. 1G), were compatible with MCTS1-DENR acting in ribosome recycling and/or re-initiation itself. Next, we confirmed that *Denr* KO in human HeLa cells indeed led to a strong downregulation of endogenous KLHDC8A protein as evaluated by immunoblot

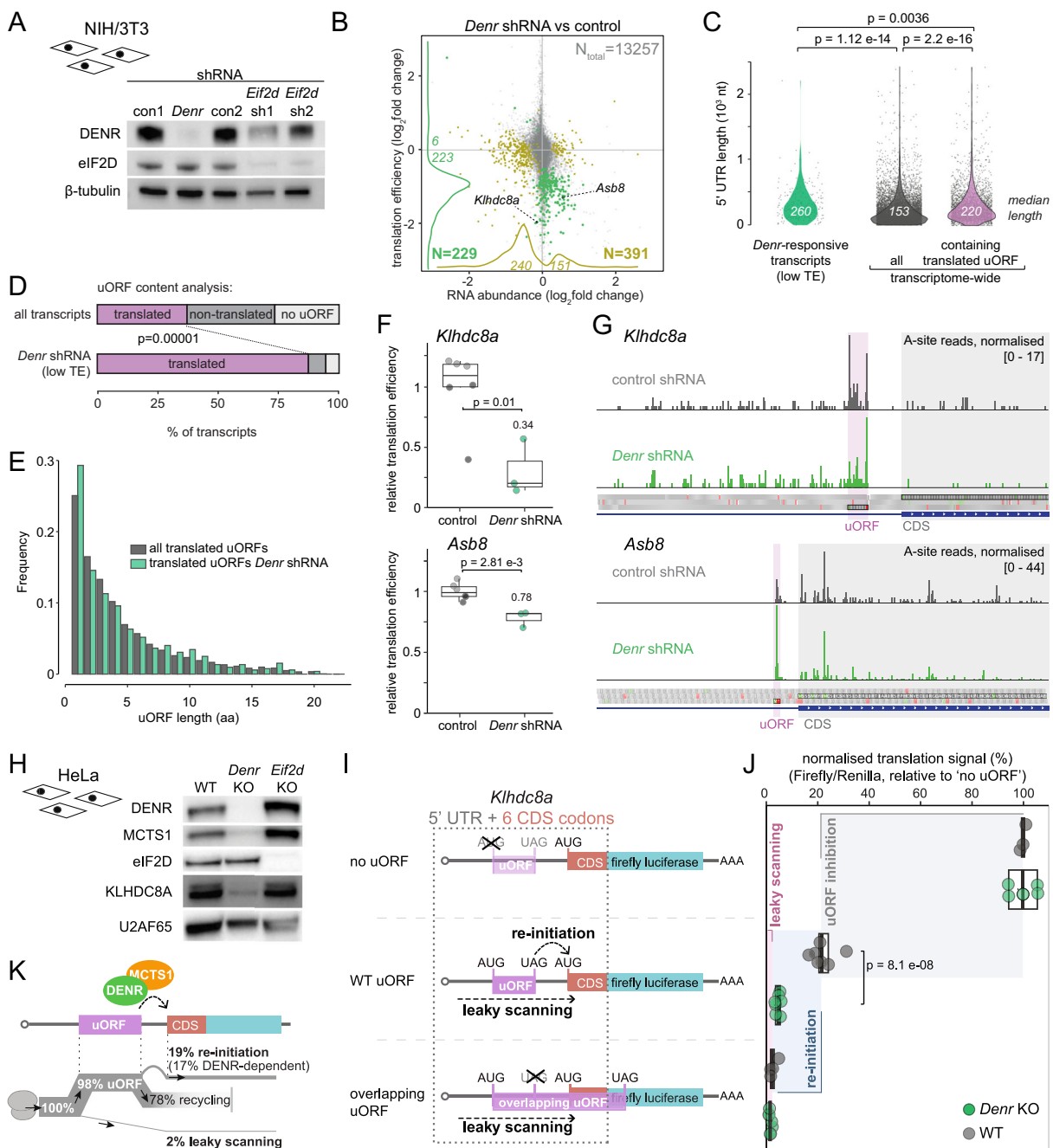

analysis using specific antibodies (Fig. 1H). The lack of functional antibodies against mouse or human ASB8 precluded a similar analysis for this protein.

We first used the *Klhdc8a* 5′ UTR (and the first 6 codons of its CDS, fused in-frame to firefly luciferase) to establish a set of three different reporter genes that would allow us to estimate re-initiation and leaky scanning rates (Fig. 1I). Thus, the *WT uORF* reporter contained the full, unmodified *Klhdc8a* 5′ UTR. Mutating the uORF start codon (*no uORF* reporter) allowed us to quantify the luciferase signal in the absence of the inhibitory uORF. A third reporter contained a mutated uORF stop codon, extending the uORF coding sequence all the way into the luciferase CDS, but in a

different frame (overlapping *uORF* reporter). Thus, this reporter would give a luciferase signal only through leaky scanning of ribosomes across the uORF start codon. After reporter delivery into HeLa cells (via lentiviral constructs that also contained a Renilla luciferase reporter for internal normalisation), re-initiation signal can be extracted from the individual reporter readouts, as shown in Fig. 1J. The relative reduction of signal from *no uORF* to *WT uORF* reporters thus represents the overall inhibitory activity of the uORF, with the remaining signal corresponding to the sum of leaky scanning- and re-initiation-mediated reporter translation. By further subtracting the *overlapping uORF* reporter signal that derives from leaky scanning, we calculate an estimate for

**Figure 1.  DENR is a re-initiation factor that promotes the translation of hundreds of uORF-containing transcripts, including *Klhdc8a* and *Asb8*.**

(A) Immunoblot analysis of NIH/3T3 cells confirms DENR and eIF2D depletion upon lentiviral transduction with *Denr* or *Eif2d* shRNAs as compared to con1 or con2 shRNAs. β-tubulin serves as a loading control. (B) Scatter plot of changes between *Denr* shRNA- and control shRNA-treated cells for RNA abundances vs. translation efficiencies (RPF counts/RNA counts), based on ribosome profiling and RNA-seq experiments. Evaluation is based on *Denr* shRNA (triplicates) and two independent control shRNAs (triplicates each). mRNAs with significant change for TE are represented in green, and mRNAs with significant change for RNA abundance are represented in olive (adjusted *p* values <0.10, Wald test followed by FDR adjustment; see Dataset EV1). Positions of *Asb8* and *Klhdc8a* are indicated. (C) Violin plots comparing 5′ UTR lengths of transcripts with lower TE upon *Denr* depletion ($n = 221$, green) vs. all expressed transcripts ($n = 9203$, grey) or the subset of all expressed transcripts that contain at least one translated uORF ($n = 3273$). *Denr*-responsive mRNAs have longer 5′ UTRs (median = 260 nt) than overall expressed transcripts (median = 153 nt) ($p = 1.12e-14$, Kolmogorov–Smirnov test), even when a comparison is restricted to the overall transcripts that also contain translated uORFs ($n = 3273$, purple; median = 220 nt; $p = 0.0036$). Of note, in this analysis, we detected for *Denr*-responsive uORF-containing transcripts on average 3.36 uORFs per UTR, and for all uORF-containing transcripts 2.28 uORFs per UTR. (D) Analysis of the proportion of transcripts with at least one translated uORF. The transcripts with lower TE upon *Denr* depletion are enriched for translated uORFs. For all expressed transcripts ($n = 9266$), 3430 carry a translated uORF, whereas 3370 have a uORF sequence that is non-translated, and 2465 have no uORF at all. In *Denr* shRNA cells, transcripts with significantly reduced TE ($n = 179$ after removal of transcripts with ambiguous/several expressed 5′ UTRs), 156 have translated uORFs vs. 13 non-translated uORFs and 10 no uORF ($p$ value <1e-5, Fisher's exact test). (E) The uORF length distribution of transcripts with reduced TE in *Denr* shRNA-transduced cells ($n = 522$) is similar to that of all expressed transcripts ($n = 7811$) ($p$ value = 0.37, Kolmogorov–Smirnov test). (F) Quantification of translation efficiencies on the coding sequence for *Klhdc8a* and *Asb8* in cells treated with shRNAs targeting *Denr* (green, $n = 3$) or controls (con1 shRNA in dark grey, $n = 3$; con2 shRNA in light grey, $n = 3$; *Klhdc8a*: $p = 0.01$ and *Asb8*: $p = 2.81e-3$, two-tailed unpaired *t*-test. The lower and upper boundaries of the boxes correspond to the first and third quartiles, while the middle line represents the median. The upper and lower whiskers extend to the largest and lowest values, limited to values that are not more distant than 1.5× the distance between the first and third quartiles. (G) Mapped footprint A-sites of *Klhdc8a* and *Asb8* transcripts in control and *Denr* knock-down cells. Read counts were normalised to library depth by subsampling, and replicates were merged for increased coverage. (H) Western Blot analysis of WT, *Denr* KO and *Eif2d* KO HeLa cells validates depletion of KLHDC8A in the absence of DENR. U2AF65 serves as loading control. (I) Schematic of the *Klhdc8a* 5′ UTR reporters used for the cellular and the in vitro translation assays. The 'no uORF reporter' serves to evaluate the regulation by the 5′ UTR in the absence of the uORF (100% signal). The 'overlapping uORF reporter' can only produce a luciferase signal through leaky scanning. Comparison of the WT reporter, containing the relevant uORF, with the two former reporters allows to specifically calculate re-initiation. Not depicted: all plasmids also express Renilla luciferase for internal normalisation. (J) Normalised luminescence signal (firefly/Renilla) of the *Klhdc8a* reporters after transduction in WT and *Denr* KO HeLa cells. The canonical initiation signal of the 'no uORF reporter' was set to 100% in WT and *Denr* KO cells individually. The shaded boxes represent the signal that can be calculated for uORF inhibition (grey; 'no uORF' signal minus 'WT uORF' signal), re-initiation (blue; 'WT uORF' signal minus 'overlapping uORF' signal) and leaky scanning (pink; 'overlapping uORF' signal) (significance calculated using two-tailed unpaired *t*-test). WT, no uORF: $n = 3$; *Denr* KO, no uORF: $n = 5$; WT, uORF: $n = 7$; *Denr* KO, uORF: $n = 7$; WT, overlapping uORF: $n = 4$; *Denr* KO, overlapping uORF: $n = 7$. The lower and upper boundaries of the boxes correspond to the first and third quartiles, while the middle line represents the median. The upper and lower whiskers extend to the largest and lowest values, limited to values that are not more distant than 1.5× the distance between the first and third quartiles. (K) Schematic representation of ribosomal fluxes on *Klhdc8a* 5′ UTR estimated from the results shown in panel (J). Source data are available online for this figure.

re-initiation. Using WT and *Denr* KO HeLa cells (Fig. 1H), this assay indicated strong inhibition by the *Klhdc8a* uORF, and the remaining signal that was largely DENR-dependent, as well as very low leaky scanning rates (Fig. 1J). The data further allowed us to propose a model of how ribosomal fluxes occur on the *Klhdc8a* reporter transcript in HeLa cells (Fig. 1K). Our analyses suggest that ~98% of ribosomes initiate on the uORF, and only ~2% undergo leaky scanning. Assuming further that ribosomes committed to uORF translation can undergo the two fates after termination—recycling or CDS re-initiation—and that there is no interference between uORF ribosomes, we would estimate that about 19% re-initiate onto the main CDS, a process that is largely DENR-dependent. Based on considerations that we further develop in the Discussion, we believe that this estimate of re-initiation probability may further represent a lower limit for re-initiation, with actual rates quite possibly higher.

We concluded that the translation of endogenous and reporter-encoded KLHDC8A was highly re-initiation-dependent and that our system based on three reporters, together with the use of WT vs. *Denr* KO cells, allowed the quantification of DENR-dependent re-initiation in vivo.

## An in vitro re-initiation assay recapitulates DENR- and uORF-dependent regulation observed in vivo

Next, we set out to establish an in vitro assay that could recapitulate re-initiation. To this end, we prepared translation-competent extracts from HeLa cells, essentially adapting a recently reported protocol based on detergent-free cell lysis applying reproducible shearing forces via a dual centrifugation step (Gurzeler et al, 2022)

(Fig. 2A). Under the gentle lysis conditions used, a sizeable proportion of cells remained intact (Fig. 2B, pellet fraction), yet we found the supernatant to reproducibly recover highly translation-competent lysate that also contained the three proteins of interest for our study (MCTS1, DENR and eIF2D) (Fig. 2B). Initial tests revealed that during in vitro translation reactions in our extracts, eIF2α became phosphorylated at Ser51 (Fig. 2B), a modification that is associated with decreased initiation rates leading to low translational output. As previously reported (Gurzeler et al, 2022; Mikami et al, 2010), the addition of recombinant GADD34Δ1-240 (a protein that recruits protein phosphatase 1 to eIF2α to induce its dephosphorylation) greatly counteracted Ser51 phosphorylation (Fig. 2C). Indeed, in the absence of GADD34Δ1-240, luciferase signals that were obtained in in vitro translation assays for in vitro transcribed, capped and polyadenylated reporter RNAs were >50% reduced as compared to reporter signals in the presence of recombinant GADD34Δ1-240 (Fig. 2D). As default condition in our experiments, we hence added recombinant GADD34Δ1-240 to all reactions. Finally, we further optimised the *Klhdc8a*-based reporters by shortening the 5′ UTR from its original ~650 nt down to ~180 nt, retaining the larger uORF environment yet removing far upstream 5′ UTR sequence (Appendix Fig. S2A). These short *Klhdc8a* 5′ UTR reporters showed strongly increased signals in our assays (Appendix Fig. S2B), in line with the notion that 5′ UTR length generally anticorrelates with CDS translation efficiency (e.g. (Bohlen et al, 2020a; Janich et al, 2015)).

First, we carried out the in vitro translation assays on the set of *Klhdc8a* reporters (no uORF, WT uORF, overlapping uORF), using HeLa extracts prepared from both wild-type and *Denr* KO cells. These experiments revealed clear uORF- and DENR-dependence

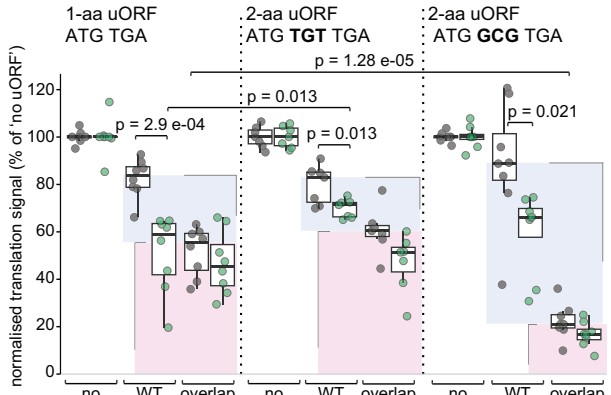

**Figure 2. In vitro translation of *Klhdc8a* and *Asb8* reporters recapitulates regulation in vivo.**

(A) Outline of the preparation of HeLa translation-competent extracts and of the capped RNA reporters for the in vitro translation assays. The Renilla reporter mRNA was used throughout in vitro assays for normalisation purposes. (B) Immunoblot analysis of proteins extracted from HeLa cells, i.e. whole cell lysate, supernatant and pellet post-dual-centrifugation and translation-competent extract post-translation. MCTS1, DENR, eIF2D and RPL23 are consistently recovered in the translation-competent extracts. During the in vitro translation reaction, eIF2α-Ser51 becomes phosphorylated. (C) Immunoblot analysis of whole HeLa cell lysate and translation-competent extracts after in vitro translation reveals that the increase in eIF2α-p51 occurring during the in vitro translation reaction can be prevented by the addition of recombinant GADD34Δ1-240. β-tubulin serves as a loading control. (D) Raw luminescence signals of firefly and Renilla luciferase reporters after in vitro translation in HeLa translation-competent extracts complemented with or without GADD34Δ1-240 validates reduced translational output upon eIF2α phosphorylation. $n = 17$ for each condition. (E) Normalised luminescence signal (firefly/Renilla) of the *Klhdc8a* reporters after in vitro translation in WT and *Denr* KO HeLa lysates (left) and rescue assay with 0.5 μM recombinant MCTS1-DENR (right); 'no uORF reporter' signal was set to 100% and uORF inhibition, re-initiation and leaky scanning were calculated as indicated by the shaded boxes and as in Fig. 1J (significance between 'WT uORF' signal in WT HeLa lysate and 'WT uORF' signal in *Denr* KO lysate calculated using two-tailed unpaired *t*-test). All non-rescue conditions (left part of plot) $n = 6$–7; all rescue conditions (right part of plot): $n = 5$. The lower and upper boundaries of the boxes correspond to the first and third quartiles, while the middle line represents the median. The upper and lower whiskers extend to the largest and lowest values, limited to values that are not more distant than 1.5× the distance between the first and third quartiles. (F) Schematic representation of ribosomal fluxes on the *Klhdc8a* 5′ UTR reporter estimated from the results shown in panel (E). (G) As in (E), but for the *Asb8* reporters. All non-rescue conditions (left side of plot): $n = 6$; all rescue conditions (right side of plot): $n = 5$. (H) As in (F) for the *Asb8* 5′ UTR in vitro data. (I) Normalised luminescence signal (firefly/Renilla) of the *Klhdc8a* reporters with mutated uORF penultimate codon (GTG to GTC or GCG) reveals the effect of penultimate codon identity on DENR-dependence. Significance was calculated using a two-tailed unpaired *t*-test. All wild-type constructs (left part of plot): $n = 5$–6. All mutant constructs (middle and right part of plot): $n = 7$–9. The lower and upper boundaries of the boxes correspond to the first and third quartiles, while the middle line represents the median. The upper and lower whiskers extend to the largest and lowest values, limited to values that are not more distant than 1.5× the distance between the first and third quartiles. (J) Normalised luminescence signal (firefly/Renilla) of the *Asb8* reporters with an inserted codon (TGT or GCG) between the uORF start and stop codons indicates that lengthening of the uORF decreases DENR-dependence. Pairwise significances were calculated using a two-tailed unpaired *t*-test. All wild-type and GCG mutant conditions: $n = 8$; all TGT mutant conditions: $n = 7$. Two-way-ANOVA revealed a significant interaction between genotype and reporter length for the 1-aa vs. TGT reporter ($p = 0.022$), indicating that the difference between wild-type and *Denr* KO lysates was significantly different between the 1-aa and 2-aa uORF constructs. The lower and upper boundaries of the boxes correspond to the first and third quartiles, while the middle line represents the median. The upper and lower whiskers extend to the largest and lowest values, limited to values that are not more distant than 1.5× the distance between the first and third quartiles. Source data are available online for this figure.

(Fig. 2E, left side; identical outcomes were seen for 'long *Klhdc8a* 5′ UTR' reporter, see Appendix Fig. S2C), closely resembling our observations from HeLa cells in vivo (Fig. 1J). The calculated ribosomal fluxes showed excellent correspondence as well (Fig. 2F). In particular, the uORF strongly impeded CDS expression with a very low rate of leaky scanning (1%). Re-initiation-mediated reporter expression (11%) was quantitatively lower than in vivo (19%, see Fig. 1K), yet remained highly DENR-dependent (i.e. in *Denr* KO extracts, the signal from the *WT uORF* reporter was similar to the leaky scanning signal from the *overlapping uORF* reporter; a summary of calculated re-initiation rates from all in vitro experiments of this study is also given in Appendix Fig. S3). Importantly, complementation with recombinant MCTS1-DENR (Appendix Fig. S2D,E) fully rescued re-initiation in *Denr* KO extracts (Fig. 2E, right), confirming the direct involvement of these proteins in re-initiation in vitro. Note also that in *Denr* KO, MCTS1 is strongly co-depleted (Fig. 1H) as a result of its instability in the absence of its binding partner (as previously observed in (Ahmed et al, 2018; Bohlen et al, 2020b)), making a rescue with both proteins necessary. We concluded that our in vitro assays showed very good correspondence with re-initiation parameters observed in HeLa cells in vivo.

Next, we assayed in vitro the start-stop uORF-containing *Asb8* reporter set (Fig. 2G). These experiments revealed that this uORF was rather permissive to leaky scanning, likely due to poor Kozak context of its start codon; in particular, given that all stop codons (TGA, TAA, TAG) begin with a T nucleotide, start-stop uORFs by default have a (suboptimal) T in Kozak sequence +4 position. The lower efficiency of inhibition by the *Asb8* uORF (as compared to the *Klhdc8a* uORF) was in line with the translation efficiencies measured in cells by Ribo-seq (Fig. 1F). According to our calculations (Fig. 2H), 58% of ribosomes underwent leaky scanning and 42% engaged in uORF translation. About two-thirds of uORF termination events would lead to re-initiation at the CDS and the

remaining one-third would not (29 vs. 13%). Re-initiation was mostly MCTS1-DENR-dependent, as judged from the signals obtained in *Denr* KO extracts and from recombinant MCTS1-DENR rescue (Fig. 2G; Appendix Fig. S3).

MCTS1-DENR selectivity for certain uORFs has been linked to specific penultimate codons (Bohlen et al, 2020b). The *Klhdc8a* uORF terminates with the sequence '… GTG(Val) TAG(Stop)', yet penultimate GTG was not observed as specifically associated with DENR-dependence previously (Bohlen et al, 2020b). In order to test penultimate codon influence in our assay, we tested reporters in which the GTG(Val) was mutated to GTC(Val) (among the reported significantly depleted penultimate codons in DENR-dependent uORFs (Bohlen et al, 2020b)) and to GCG(Ala) (most strongly and significantly enriched in DENR-dependent uORFs according to (Bohlen et al, 2020b)) (Fig. 2I). We found that both the inhibitory impact of the uORF and the re-initiation rates were only weakly affected by these codon changes in vitro. We noted that in the depleted penultimate codon case, GTC(Val), DENR-dependence of re-initiation appeared somewhat reduced, yet overall re-initiation capacity was still intact. This observation is compatible with the idea that the depleted penultimate codon is able to partially decouple the re-initiation process from its dependence on these particular re-initiation factors. Moreover, it may point to the involvement of MCTS1-DENR in the process of specific deacylated tRNA removal.

Next, we examined *Asb8* reporter variants, converting the 1-aa uORF to a 2-aa uORF (and thus from start-stop to elongating uORF) and testing different penultimate codons. We inserted either a TGT(Cys), which is (according to (Bohlen et al, 2020b)) the only significantly enriched penultimate codon starting with 'T' (hence preserving the +4 Kozak context of the start codon), or GCG(Ala), which is a highly enriched codon and additionally improves the Kozak context for uORF initiation (Fig. 2J). In the in vitro translation assays, GCG insertion indeed led to strongly decreased

leaky scanning rates, converting the weakly translated 1-aa uORF to a strongly translated 2-aa uORF (Fig. 2J, right), compatible with the improved Kozak consensus associated with the +4 position 'G'. Lengthening the uORF with TGT(Cys) had little consequences on leaky scanning and overall re-initiation rate, yet DENR-dependence was significantly reduced (Fig. 2J, middle). We concluded that re-initiation rates were not per se different between start-stop and elongating uORFs, but the DENR-dependence of re-initiation was modulated, which is in agreement with the reported preferential MCTS1-DENR regulation of start-stop uORFs (Bohlen et al, 2020b; Schleich et al, 2017).

## Phospho-eIF2α levels only mildly influence re-initiation rates of *Klhdc8a* and *Asb8* reporters

The phosphorylation of eIF2α plays a crucial role in the integrated stress response (ISR) by reducing overall translation initiation and selectively enhancing the translation of stress-related mRNAs. This process operates through a mechanism involving uORFs, re-initiation, and start codon selection (Dever et al, 2023). Following the termination of uORF translation and partial ribosome recycling —specifically the dissociation of the 60S subunit and removal of deacylated P-site tRNA—the 40S ribosomal subunit regains its scanning competence and can eventually re-initiate translation at a downstream start codon. According to the 'delayed translation re-initiation model', which is based on studies of transcripts such as *Atf4/Gcn4*, in conditions where non-phosphorylated eIF2α is abundant, a new eIF2-GTP-Met-tRNA$^{Met}_i$ complex (also known as ternary complex, TC) is quickly recruited during the 40S scanning process. This rapid acquisition leads to translation initiation at a nearby second uORF (87 nt away) rather than at the more distant CDS, which is 184 nt downstream. Phosphory-lated eIF2α inhibits the guanine nucleotide exchange factor eIF2B, which is necessary to convert inactive GDP-bound eIF2α to its active GTP-bound form, thereby limiting the availability of TC. In this scenario, the likelihood of timely TC recruitment for translation of the second uORF is diminished, increasing the chances of re-initiation at the downstream CDS. Recent studies have underscored the crucial roles of MCTS1-DENR and eIF2D in facilitating *Atf4* CDS translation (Bohlen et al, 2020b; Vasudevan et al, 2020); however, the specific stage at which these proteins exert their effects— whether during ribosome recycling, TC recruitment, or re-initiation itself—remains unclear. Intriguingly, eIF2D was originally identified as having the ability to bind initiator tRNA in a GTP-independent fashion and deliver Met-tRNA$^{Met}_i$ to the P-site in vitro (Dmitriev et al, 2010). This has led to the hypothesis that MCTS1-DENR and eIF2D might perform a similar function in vivo, potentially making TC activity for initiator tRNA delivery redundant during re-initiation—a model that has seen some critical review recently (e.g. (Grove et al, 2024)), prompting the question of whether TC/eIF2α activity is a limiting factor in MCTS1-DENR-mediated re-initiation, independent of its effects on global initiation.

We reasoned that our well-controlled in vitro system would allow for dedicated investigations in this direction. We thus carried out our assays in the presence vs. absence of GADD34Δ1-240, which led to low and high Ser51 phosphorylation levels of eIF2α, respectively (Fig. 2C). Of note, the level of Ser51 phosphorylation occurring in the absence of GADD34Δ1-240 in vitro was even

higher than that seen in vivo after a standard ISR induction protocol by tunicamycin treatment of cells (Appendix Fig. S4A). For the *Klhdc8a* reporters, absolute signals were significantly reduced in the absence of GADD34Δ1-240 as expected (analogous to experiments in Fig. 2D), yet after normalisation to the (likewise reduced) Renilla luciferase signal, there were not significant changes to the estimated levels of re-initiation (Fig. 3A,B; Appendix Fig. S3). We concluded that in the in vitro experiments using the *Klhdc8a* reporters, the availability of non-phosphorylated eIF2α was not strongly rate-limiting for the overall re-initiation process. On the *Asb8* reporter—where the intercistronic distance between uORF and CDS is very short (17 nt, as compared to 73 nt in *Klhdc8a*)—we found overall more variability in the experimental outcomes in the absence of GADD34Δ1-240 (Fig. 3C, D). Thus, we observed a tendency to increased signal from leaky scanning and of reduced signal attributable to re-initiation (both non-significant). In an in vivo experiment on the *Klhdc8a* reporter set, combined with tunicamycin treatment, we also observed only a weak reduction in re-initiation rate (Appendix Fig. S4B, C) and validated that under these conditions, the translation of positive control reporters carrying the *Atf4* or *Atf5* 5′ UTRs were indeed upregulated (Appendix Fig. S4D). Taken together, these results suggest that eIF2α activity is only moderately limiting for re-initiation per se, as judged by our 'single uORF' reporters, *Klhdc8a* and *Asb8*. One may speculate that it is, in particular, in the context of 'multiple uORF' configurations and competing re-initiation events—i.e. in cases such as on *Atf4* and *Atf5*— that the influence of phospho-eIF2α comes to bear. Our results can serve as a starting point for future investigations into possible differential TC requirements across different uORF-dependent re-initiation events.

## Limited evidence for the role of eIF2D in re-initiation in vivo and in vitro points to uORF-independent cellular functions in gene expression regulation

Next, we extended our analyses to the close MCTS1-DENR homologue, eIF2D, which contains the same domains encoded as a single polypeptide, linked by an additional WH domain (Fig. 4A). The high similarity between the proteins, in addition to the findings that the yeast orthologues show shared activities, has led to the assumption that the homology in mammals extends to their function. To test this hypothesis, we downregulated eIF2D using specific shRNAs (Fig. 1A) and carried out Ribo-seq and RNA-seq identically to the experiment in *Denr*-depleted cells shown in Fig. 1B. For a first comparison between the two genotypes, i.e. *Eif2d*-depleted and *Denr*-depleted, we evaluated in a transcriptome-wide fashion the relative ribosome distribution between mRNA 5′ UTR and CDS sequences. This measure is expected to increase when a re-initiation factor is depleted because ribosomes are selectively lost from CDS sequences but will still translate uORFs when re-initiation is inhibited (Castelo-Szekely et al, 2019). Indeed, the distribution of the changes seen for 5′ UTR-to-CDS footprint ratios between *Denr* shRNA-treated cells vs. control cells was strongly shifted to the positive (Fig. 4B), indicating globally detectable ribosome redistribution in the absence of MCTS1-DENR consistent with previous findings (Castelo-Szekely et al, 2019). By contrast, the shift was considerably smaller in the cells that had been treated with *Eif2d* shRNAs (Fig. 4B). Next, we carried out an analysis identical to that applied to *Denr* knockdown cells in

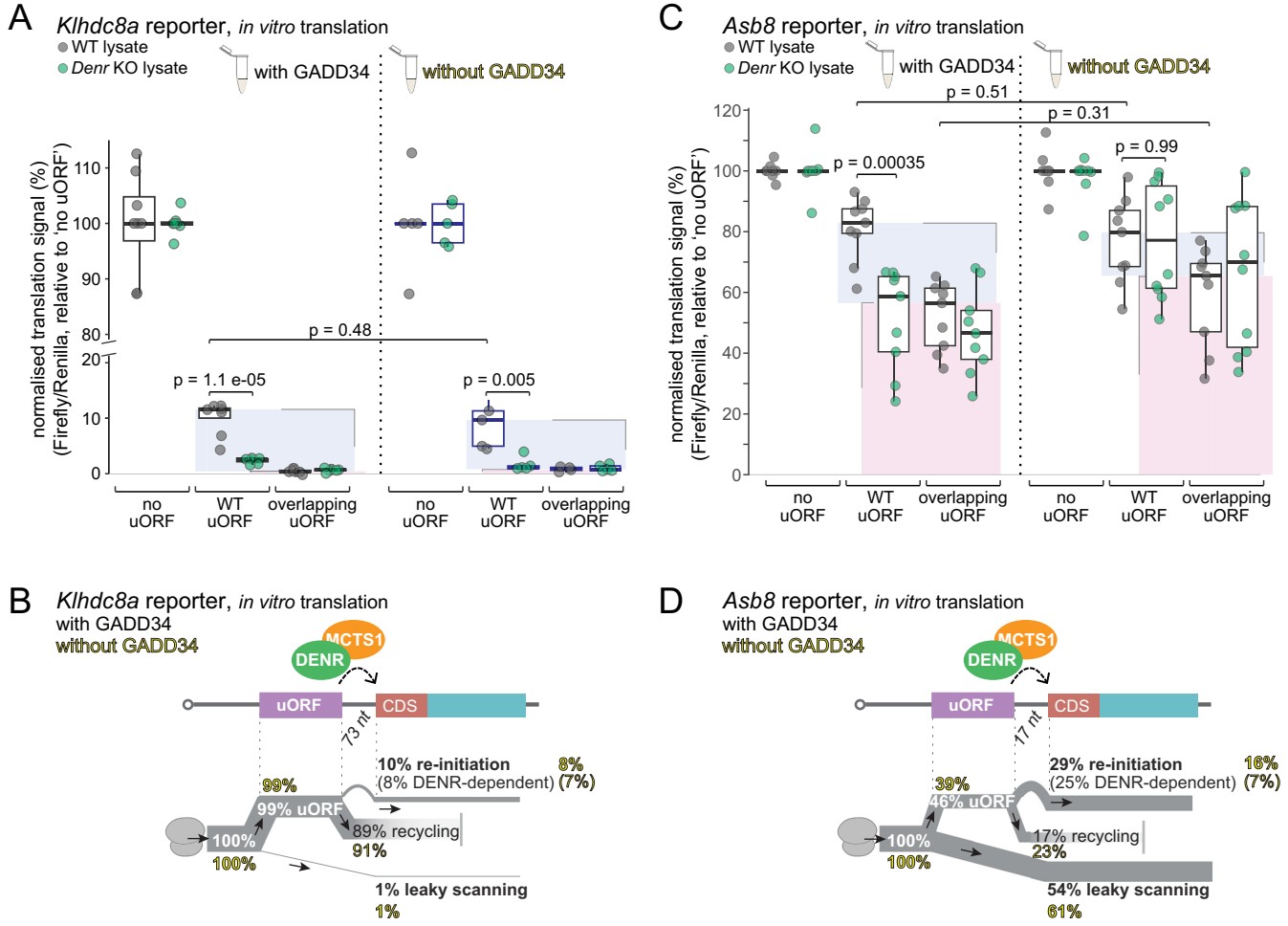

**Figure 3. eIF2α phosphorylation status moderately affects re-initiation rate of start-stop uORF and longer uORF reporters.**

(A) Normalised luminescence signal of *Klhdc8a* reporters after in vitro translation in WT and *Denr* KO HeLa extracts complemented with 16 ng/µl recombinant GADD34Δ1-240 (left) or without GADD34Δ1-240 (right) (significance calculated using two-tailed unpaired *t*-test). All 'with GADD34' conditions (left part of plot): $n = 7$–8. All 'no GADD34' conditions (right part of plot): $n = 4$–5. The lower and upper boundaries of the boxes correspond to the first and third quartiles, while the middle line represents the median. The upper and lower whiskers extend to the largest and lowest values, limited to values that are not more distant than 1.5× the distance between the first and third quartiles. (B) Schematic representation of ribosomal fluxes on *Klhdc8a* 5′ UTR in vitro estimated from the results shown in panel (A). Quantifications in the presence of GADD34Δ1-240 are shown in black and without GADD34Δ1-240 in yellow. (C) As in (A) for *Asb8* reporters. For all conditions $n = 9$–10. (D) As in (B) for *Asb8* reporter data shown in (C). Source data are available online for this figure.

Fig. 1B, to determine the effects induced by *Eif2d* depletion at the level of CDS translation efficiencies (Fig. 4C). This analysis, too, revealed strikingly different transcriptome-wide effects between the homologues. We thus detected much fewer transcripts with a decreased TE upon *Eif2d* knockdown ($n = 78$, with FDR-corrected $p < 0.1$; Dataset EV2) (Fig. 4C) as compared to *Denr* knockdown cells ($n = 223$) (Fig. 1B), and the overlap between the two sets was minimal (Appendix Fig. S5A,B). Moreover, the mRNAs with low TE in *Eif2d*-depleted cells tended to have short 5′ UTRs (Appendix Fig. S5C), and they were not enriched for translated uORFs (Fig. 4D). We also noted that many of the translationally affected transcripts were highly abundant, including >25 mRNAs encoding for ribosomal proteins and general translation machinery (Appendix Fig. S5D,E). Particularly strong consequences of eIF2D loss-of-function were evident for RNA abundances, with a large number of mRNAs significantly up ($n = 460$) and downregulated ($n = 617$)

(Fig. 4C) and, here, abundant transcripts were overrepresented as well (Appendix Fig. S5F) and associated with cellular processes such as the cell cycle and translation (Appendix Fig. S5G,H). Alterations to the cell cycle could indeed be further validated as a robust phenotype following eIF2D loss-of-function (Appendix Fig. S5I). Taken together, these findings suggested that, first, the knockdown of *Eif2d* was phenotypically highly consequential as judged by the widespread gene expression reprogramming that it induced. However, second, it would seem that the primary activity of eIF2D may not lie in regulating the re-initiation of uORF-containing transcripts.

To further investigate the above observations on potential eIF2D targets, we selected several candidate transcripts to test if their TE regulation was 5′ UTR-mediated. Of these, *Cenpa*, *Ndc80*, *Med23* and *Rps20* were picked as candidate mRNAs with low TE from Fig. 4C. We also included *Hoxa3*, which has been identified as a

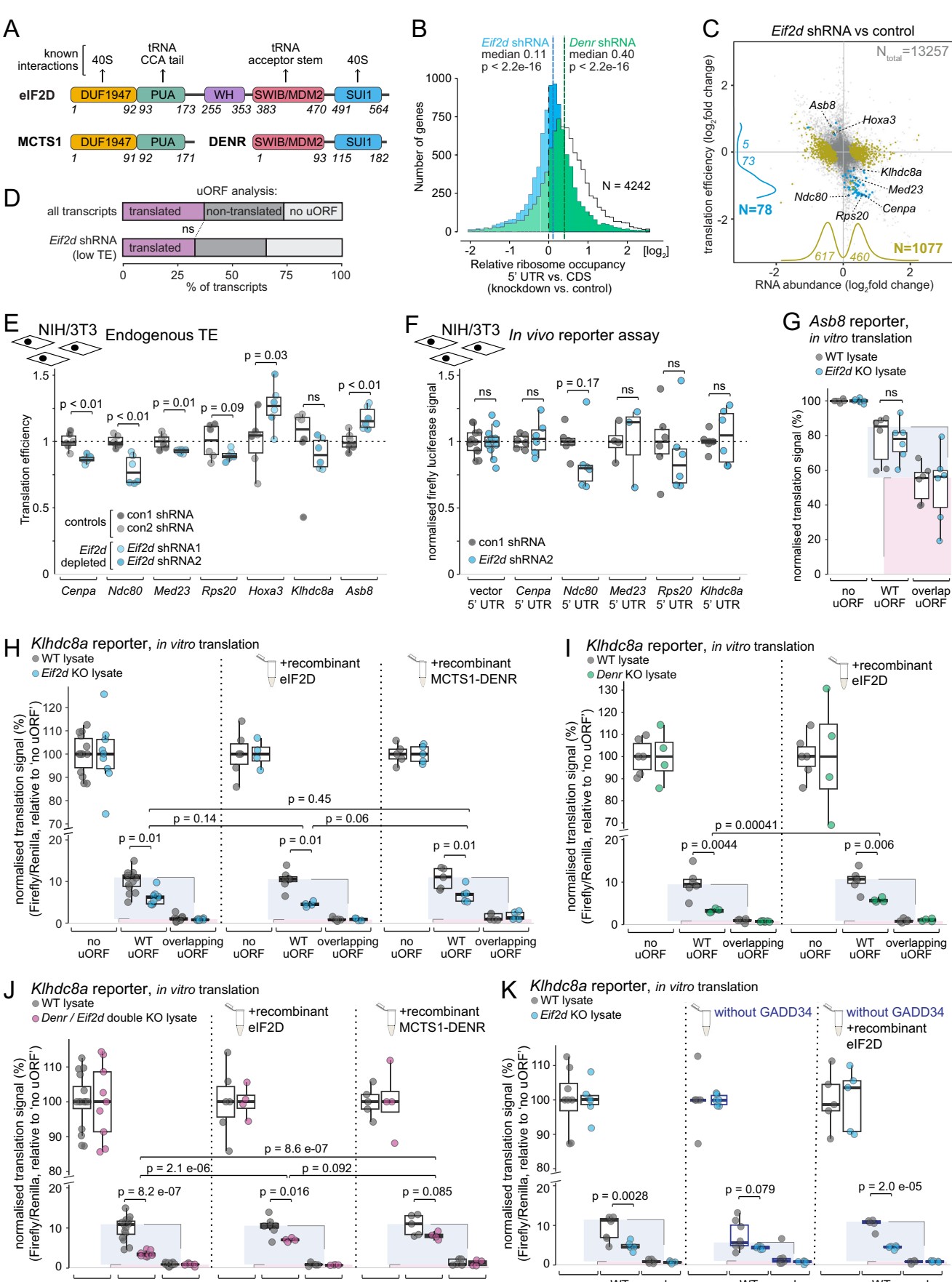

**Figure 4.   No evidence for eIF2D uORF re-initiation activity in vivo but activity in vitro.**

(**A**) Schematic representation of human MCTS1-DENR and eIF2D proteins, with domains indicated. The relevant amino acid positions defining the domains are specified. (**B**) Distribution of ribosomes on 5′ UTRs relative to CDS in *Denr* knock-down (in green) or *Eif2d* knock-down (in blue) vs. control cells ($n = 4242$). *Denr* knock-down cells show a strong redistribution of ribosomes from CDS to 5′ UTR, whereas the effect is milder but still significant for *Eif2d* knock-down cells (*p* values calculated using Wilcoxon signed rank test). (**C**) Scatter plot of changes between *Eif2d* shRNA and control cells for RNA abundances vs. translation efficiencies (RPF counts/RNA counts), based on ribosome profiling and RNA-seq experiments. Evaluation is based on two independent *Eif2d* shRNAs (triplicates each) and two independent control shRNAs (triplicates each). mRNAs with a significant change in TE were represented in blue and mRNAs with a significant change in RNA abundance were represented in olive green (adjusted *p* values <0.10, Wald test followed by FDR adjustment; see Dataset EV2). Positions of *Asb8*, *Klhdc8a*, *Hoxa3*, *Med23*, *Rps20*, *Ndc80* and *Cenpa* are indicated. (**D**) Analysis of the proportion of transcripts with at least one translated uORF. The transcripts with lower TE upon *Eif2d* depletion are not enriched for translated uORFs. For all expressed transcripts ($n = 9266$), 3430 carry a translated uORF, whereas 3370 have a uORF sequence that is non-translated, and 2465 have no uORF at all. In *Eif2d* shRNA cells, transcripts with significantly reduced TE ($n = 58$ after removal of transcripts with ambiguous/several expressed 5′ UTRs), 19 have translated uORFs vs. 19 non-translated uORFs and 20 no uORF (*p* value = 0.59, Fisher's exact test). (**E**) Quantification of translation efficiencies on the coding sequence of selected transcripts with lower TE upon *Eif2d* depletion in control and *Eif2d*-depleted cells (*p* values calculated using two-tailed unpaired *t*-test). For each of the conditions, con1 shRNA, con2 shRNA, *Eif2d* shRNA1, *Eif2d* shRNA2: $n = 3$. The lower and upper boundaries of the boxes correspond to the first and third quartiles, while the middle line represents the median. The upper and lower whiskers extend to the largest and lowest values, limited to values that are not more distant than 1.5× the distance between the first and third quartiles. (**F**) Normalised luminescence signal (firefly/Renilla) of lentivirally transduced reporters with 5′ UTRs of selected transcripts with lower TE upon *Eif2d* depletion (significance calculated using two-tailed unpaired *t*-test). Vector 5′ UTR: $n = 12$ for both control and *Eif2d* shRNA2. *Cenpa*, Ndc80m *Rps20*, *Klhdc8a* 5′ UTRs: $n = 6$ per shRNA condition. *Med23* 5′ UTR: $n = 4$ for control shRNA and $n = 3$ for *Eif2d* shRNA2. The signal of individual reporter expressed in WT cells was set to 100%. The lower and upper boundaries of the boxes correspond to the first and third quartiles, while the middle line represents the median. The upper and lower whiskers extend to the largest and lowest values, limited to values that are not more distant than 1.5× the distance between the first and third quartiles. (**G**) Normalised luminescence signal (firefly/Renilla) of the *Asb8* reporters after in vitro translation in WT and *Eif2d* KO HeLa extracts (*p* value calculated using two-tailed unpaired *t*-test). $n = 6$ for each condition. The lower and upper boundaries of the boxes correspond to the first and third quartiles, while the middle line represents the median. The upper and lower whiskers extend to the largest and lowest values, limited to values that are not more distant than 1.5× the distance between the first and third quartiles. (**H**) Normalised luminescence signal (firefly/Renilla) of the *Klhdc8a* reporters after in vitro translation in WT and *Eif2d* KO HeLa extracts (left), supplemented with 0.5 µM recombinant eIF2D (middle) or supplemented with 0.5 µM of recombinant MCTS1-DENR (right); *p* values calculated using two-tailed unpaired *t*-test. For non-rescue conditions (left part of plot): $n = 14$ for the WT conditions and $n = 9$ for the *Eif2d* KO conditions. For eIF2D rescue conditions (middle part of plot): $n = 4$–$7$. For MCTS1-DENR rescue conditions (right part of plot): $n = 5$. The lower and upper boundaries of the boxes correspond to the first and third quartiles, while the middle line represents the median. The upper and lower whiskers extend to the largest and lowest values, limited to values that are not more distant than 1.5× the distance between the first and third quartiles. (**I**) Normalised luminescence signal (firefly/Renilla) of the *Klhdc8a* reporters after in vitro translation in WT and *Denr* KO HeLa extracts (left) or supplemented with 0.5 µM recombinant eIF2D (right); *p* value calculated using two-tailed unpaired *t*-test. For all WT conditions $n = 6$; for all *Denr* KO conditions: $n = 4$. The lower and upper boundaries of the boxes correspond to the first and third quartiles, while the middle line represents the median. The upper and lower whiskers extend to the largest and lowest values, limited to values that are not more distant than 1.5× the distance between the first and third quartiles. (**J**) Normalised luminescence signal (firefly/Renilla) of the *Klhdc8a* reporters after in vitro translation in WT and *Denr* / *Eif2d* double KO HeLa extracts (left), supplemented with 0.5 µM recombinant eIF2D (middle) or supplemented with 0.5 µM recombinant MCTS1-DENR (right); *p* values calculated using two-tailed unpaired *t*-test. For non-rescue conditions (left part of plot): $n = 16$ for WT and $n = 9$ for double KO conditions. For eIF2D rescue conditions (middle part of plot): $n = 6$–$7$ for WT and $n = 4$ for double KO conditions. For MCTS1-DENR rescue conditions (right part of plot): $n = 5$ for WT and $n = 4$ for double KO conditions. The lower and upper boundaries of the boxes correspond to the first and third quartiles, while the middle line represents the median. The upper and lower whiskers extend to the largest and lowest values, limited to values that are not more distant than 1.5× the distance between the first and third quartiles. (**K**) Normalised luminescence signal (firefly/Renilla) of the *Klhdc8a* reporters after in vitro translation in WT and *Eif2d* KO HeLa extracts supplemented with GADD34Δ1-240 (left), without GADD34Δ1-240 (middle), or without GADD34Δ1-240 and complemented with 0.5 µM eIF2D (right); *p* values calculated using two-tailed unpaired *t*-test. Left and middle part of plot: $n = 8$ for WT and $n = 6$ for *Eif2d* KO conditions. For the right part of the plot: $n = 5$ for all conditions. The lower and upper boundaries of the boxes correspond to the first and third quartiles, while the middle line represents the median. The upper and lower whiskers extend to the largest and lowest values, limited to values that are not more distant than 1.5× the distance between the first and third quartiles. Source data are available online for this figure.

bona fide eIF2D-regulated mRNA in a previous study (Alghoul et al, 2021). According to the published findings, eIF2D is implicated in uORF-mediated inhibition, rather than in re-initiation, of *Hoxa3* CDS translation. Indeed, our in vivo data in cells confirmed *Hoxa3* CDS TE up-regulation upon *Eif2d* depletion (Fig. 4E; note that *Hoxa3* is very lowly expressed in our cells and did not feature among the significantly changed TE transcripts in Fig. 4C, but the differences were statistically significant in the direct comparison of TEs shown in Fig. 4E). Luciferase reporter assays in cells using the 5′ UTRs of the selected endogenous candidate transcripts did not reveal significant regulation by eIF2D (Fig. 4F). We also analysed *Klhdc8*a (endogenous and 5′ UTR reporter), but we found no indication that eIF2D was involved in its re-initiation in cells (Fig. 4E,F). Endogenous *Asb8* showed an increase in translation efficiency upon *Eif2d* depletion (Fig. 4E), contrasting the effect seen upon *Denr* depletion.

We next tested the outcome of eIF2D loss-of-function and rescue in our in vitro assay. In lysates from *Eif2d* KO HeLa cells (Fig. 1H), no difference in re-initiation activity or leaky scanning was detectable for the *Asb8* reporter (Fig. 4G). By contrast, for the

*Klhdc8a* reporter, re-initiation rates in *Eif2d* KO lysates were reduced by about half (Fig. 4H). However, neither recombinant eIF2D nor recombinant MCTS1-DENR were able to fully rescue the low signal obtained with the WT uORF reporter in *Eif2d* KO lysates (Fig. 4H). These observations were most compatible with the model that lower re-initiation signal was an indirect effect of eIF2D loss, e.g. due to secondary effects related to the widespread gene expression reprogramming seen in Fig. 4C. Thus, we would not expect indirect effects to be rescuable in vitro, whereas a direct role in the re-initiation process itself should be rescued by adding the needed re-initiation factor as a recombinant protein.

In *Eif2d* knockout extracts, MCTS1-DENR is abundant and may mask the re-initiation function of eIF2D. To address this possibility, we carried out two experiments. First, using re-initiation-deficient *Denr* KO extracts and recombinant eIF2D, we were indeed able to partially rescue re-initiation activity (Fig. 4I), although not as efficiently as with recombinant MCTS1-DENR (compare with Fig. 2E). Second, we used *Denr* + *Eif2d* HeLa double knockout cell lysates. In vitro translation reactions in these extracts showed reduced re-initiation activity (Fig. 4J, left) that was very

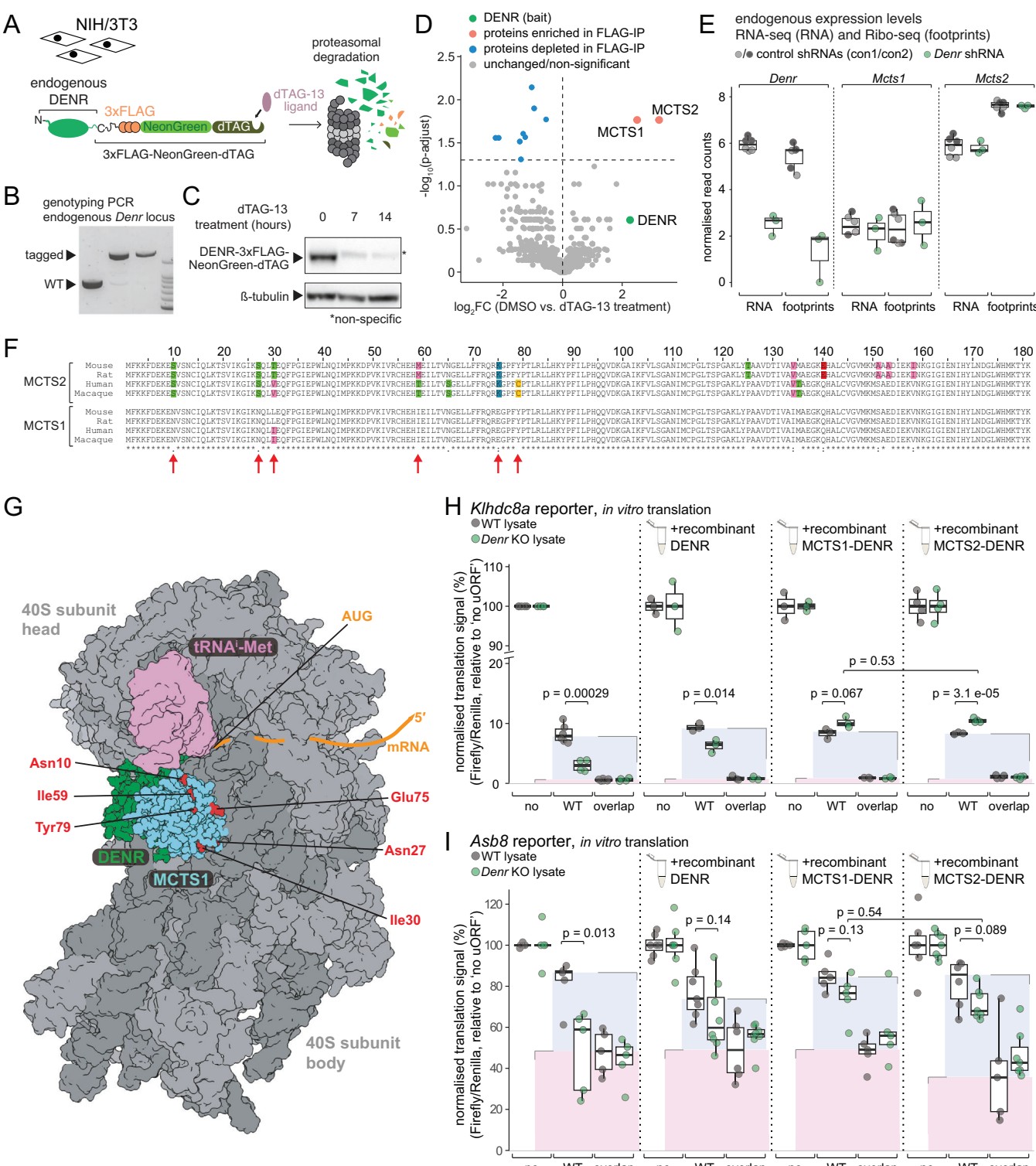

similar to *Denr* single knockouts (Fig. 2E). Addition of recombinant MCTS1-DENR (0.5 µM) showed the expected rescue (Fig. 4J, right). Interestingly, also recombinant eIF2D (0.5 µM) showed rescue of the double knockout (Fig. 4J, middle). Although rescue with recombinant eIF2D appeared less potent than rescue with recombinant MCTS1-DENR at first sight—i.e., in the right panel to

Fig. 4J, there is no significant difference on WT uORF reporter between MCTS1-DENR-supplemented WT vs. MCTS1-DENR-supplemented double KO lysates, whereas in the middle panel, double KO lysate supplemented with recombinant eIF2D still showed significantly lower signal than WT lysate supplemented with eIF2D—two-way ANOVA did not uncover a significant

**Figure 5. MCTS2 interacts with DENR in vivo and promotes re-initiation in vitro.**

(A) Schematic representation of endogenous DENR tagged with 3×FLAG-NeonGreen-dTAG. Treatment with the dTAG-13 ligand targets DENR-3×FLAG-NeonGreen-dTAG for proteasomal degradation. (B) Genotyping PCR analysis of *Denr*-tagged clones validates the insertion of the 3×FLAG-NeonGreen-dTAG cassette upstream of *Denr* stop codon. The tagged *Denr* amplicon has a length of 2067 bp and a WT *Denr* of 885 bp. (C) Western Blot analysis of NIH/3T3 DENR-3×FLAG-NeonGreen-dTAG cells treated for 7 or 14 h with 500 nM dTAG-13 shows efficient depletion of DENR upon treatment. An asterisk indicates a non-specific band that migrates at a similar height as the tagged protein. (D) Mass spectrometry analysis of proteins co-immunoprecipitated with endogenously tagged DENR in non-treated vs. dTAG-13-treated cells reveals the interaction of MCTS2 with DENR. Only DENR and significant hits enriched in non-treated cells are shown (*p* values calculated using a two-tailed unpaired *t*-test followed by correction for multiple testing). (E) Normalised RNA abundances and ribosome footprints of *Denr*, *Mcts1* and *Mcts2* in control and DENR-depleted NIH/3T3 cells. For con1, con2 and *Denr* shRNA: $n = 3$. The lower and upper boundaries of the boxes correspond to the first and third quartiles, while the middle line represents the median. The upper and lower whiskers extend to the largest and lowest values, limited to values that are not more distant than 1.5× the distance between the first and third quartiles. (F) Alignment of human, rhesus macaque, mouse and rat MCTS1 and MCTS2 amino acid sequences. The amino acids changing relative to mouse MCTS1 are highlighted with different colours according to their side-chain chemistry. (G) Structural model of MCTS1-DENR (turquoise/green) in interaction with the 40S ribosomal subunit (grey), Met-tRNA$^{Met}_i$ (magenta) and an mRNA that is shown schematically on its predicted trajectory (orange). The amino acids that undergo changes in MCTS2 are indicated in red in the structure and labelled. PDB accession numbers used for depiction: 5vyc (40S and MCTS1-DENR), 5oa3 (initiator tRNA), 6ms4 (MCTS1-interacting domain of DENR). (H) Normalised luminescence signal (firefly/Renilla) of the *Klhdc8a* reporters after in vitro translation in WT and *Denr* KO HeLa lysates (left), supplemented with 0.5 μM of recombinant DENR, MCTS1-DENR or MCTS2-DENR as indicated (*p* values calculated using two-tailed unpaired *t*-test). No rescue conditions (left): $n = 6$ for WT and $n = 4$ for *Denr* KO. DENR and DENR-MCTS1 rescue conditions (middle): $n = 3$ throughout. DENR-MCTS2 rescue conditions (right): $n = 4$ throughout. The lower and upper boundaries of the boxes correspond to the first and third quartiles, while the middle line represents the median. The upper and lower whiskers extend to the largest and lowest values, limited to values that are not more distant than 1.5× the distance between the first and third quartiles. (I) As in (H), for the *Asb8* reporters. No rescue conditions (left): $n = 5$ throughout. DENR rescue conditions: $n = 6–8$. DENR-MCTS1 rescue conditions: $n = 5$ throughout. DENR-MCTS2 rescue conditions (right): $n = 5–7$. The lower and upper boundaries of the boxes correspond to the first and third quartiles, while the middle line represents the median. The upper and lower whiskers extend to the largest and lowest values, limited to values that are not more distant than 1.5× the distance between the first and third quartiles. Source data are available online for this figure.

interaction and we concluded that eIF2D can potentially rescue as potently as MCTS1-DENR. Taken together, these results indicate that eIF2D is likely not an essential component of the re-initiation machinery in cells in vivo and in vitro under wild-type conditions where MCTS1-DENR is abundantly present. However, when MCTS1-DENR is absent, providing eIF2D protein can compensate for MCTS1-DENR loss. Thus, it is a probable scenario that eIF2D has, in principle, retained shared activity with MCTS1-DENR, but we deem it likely that its main cellular functions under physiological conditions lie outside of uORF re-initiation.

Finally, it is also possible that eIF2D is active in re-initiation in a conditional fashion; in particular, upon induction of the integrated stress response that leads to the unavailability of eIF2α, eIF2D may be able to deliver initiator-tRNA to the 40S ribosome. We addressed this possibility by carrying out in vitro translation in *Eif2d* knockout extracts (without/with recombinant eIF2D rescue) in the absence of GADD34Δ1-240 (Fig. 4K). Also, under these conditions, we observed the (presumed) indirect/secondary effects of eIF2D loss that could not be rescued by recombinant eIF2D, and the experiment revealed no further indications for a specific re-initiation role of eIF2D.

## Identification of MCTS2 as a functional DENR heterodimerisation partner

It has been observed that MCTS1 and DENR are mutually dependent on each other for protein stability (Ahmed et al, 2018; Bohlen et al, 2020b), giving rise to the idea that they act as obligate heterodimers. However, closer inspection of the available cell culture data would suggest that the collateral depletion of MCTS1 in *Denr* KO cells is stronger than that of DENR in *Mcts1* KOs (Bohlen et al, 2020b); moreover, in vivo phenotypes in mouse neuronal migration are stronger upon DENR loss than MCTS1 loss (Haas et al, 2016). These findings could indicate that DENR is indeed an obligatory component of the heterodimer, but that it can interact with, and be stabilised by, alternative heterodimerisation

partners as well. To obtain insights into this hypothesis, we sought to purify DENR-interacting proteins by a co-immunoprecipitation and mass-spectrometry approach. Using CRISPR gene editing, we modified the endogenous *Denr* locus in NIH/3T3 cells such that it expressed DENR protein with a C-terminal tag suitable for immunoprecipitation (3×FLAG) and degron-mediated protein depletion (dTAG/dTAG-13 degrader system (Nabet et al, 2018)) (Fig. 5A). Genotyping indicated successful modification of both alleles in selected cell clones (Fig. 5B) and immunoblot analysis validated the functionality of the degron system (Fig. 5C). We then analysed by mass-spectrometry the proteins enriched in FLAG-IP performed on extracts from control-treated vs. dTAG-13-treated cells (Fig. 5D). The two most strongly enriched co-purifying proteins were MCTS1 and MCTS2. This finding indicates that DENR assembles into two distinct heterodimeric complexes, MCTS1-DENR and MCTS2-DENR. *Mcts2* is a paralog and retrogene copy of *Mcts1* that has been noted for its transcription from an imprinted gene region in mouse and humans, with predominant expression from the paternal allele (Wood et al, 2007). A recent study on human *Mcts1* deficiency and associated disease phenotypes has speculated on redundancy with *Mcts2*, yet based on reporter assays and other analyses, it came to the conclusion that this paralog was largely non-functional (Bohlen et al, 2023). By contrast, our interaction data collected from endogenous proteins suggests that both MCTS paralogs form bona fide heterodimers with DENR. The *Mcts1* and *Mcts2* coding sequences differ at 81 individual nucleotide positions that are distributed quite evenly across the overall CDS of 546 nt (Appendix Fig. S6A), allowing to unambiguously assign RNA-seq reads and Ribo-seq footprints to the two paralogous sequences. We could thus compare the expression of the *Mcts* paralogs, which indicated higher levels of *Mcts2* than *Mcts1* in our cells (Fig. 5E). These findings strongly indicate that MCTS2 is a relevant interaction partner of DENR besides MCTS1.

Human and mouse MCTS1 protein sequences are identical apart from one leucine to isoleucine substitution at amino acid position

30 (Fig. 5F). In MCTS2, divergence from MCTS1 can be found at 14 positions when aligning mouse, rat, human and macaque protein sequences, with four substitutions common to MCTS2 across species and other changes specific to the analysed rodents or primates (Fig. 5F). Using available structural data from in vitro reconstituted MCTS1-DENR in complex with the 40S ribosome, an initiator-tRNA molecule and a viral IRES-containing model RNA (Weisser et al, 2017) (please note that no such structure has so far been reported for a natural, non-viral transcript), we found that MCTS2-specific amino acid changes affected positions that all cluster in the same region of the fold but outside of the interaction surface between MCTS1-DENR and also outside of that between MCTS1-DENR and the 40S ribosomal subunit (Fig. 5G). Hence, the capacity of MCTS2 vs. MCTS1 to interact with DENR, and of MCTS2-DENR vs. MCTS1-DENR to interact with 40S can be expected to be comparable. However, the substitutions—of which in particular the change at amino acid position 75 from glutamate (negative charge) in MCTS1 to lysine (positive charge) in MCTS2 (Fig. 5F) would have the potential to significantly alter protein properties—could affect additional, so far unknown interactions taking place at this solvent-exposed surface within the complex (Fig. 5G). To gain insights into whether MCTS1-DENR and MCTS2-DENR could redundantly act in re-initiation, we tested both as recombinant proteins on *Klhdc8a* and *Asb8* reporters. Addition of recombinant DENR alone showed only low rescue activity, likely due to residual MCTS1/2 in *Denr* KOs (see Fig. 1H), yet supplementing with either of the MCTS paralogs together with DENR led to a comparable and full rescue (Fig. 5H,I). Taken together, we concluded that *Mcts2*, although previously categorised as a pseudogene (Nandi et al, 2006), is expressed to relevant levels and produces a protein that assembles with DENR into protein complexes in cells in vivo. In vitro, the paralogous complexes show similar activity on our two re-initiation model substrates, indicating redundancy in activity, at least for certain substrates. Finally, it is noteworthy that our anti-MCTS1 antibody recognises recombinant MCTS1 and MCTS2 with similar efficiency (Appendix Fig. S6B), indicating that western blot quantifications of endogenous MCTS1 in our and other previous studies likely already reported on the combined protein levels of the two paralogs.

## Discussion

Translation re-initiation is a process that is at odds with several principles of canonical initiation, termination and post-termination subunit recycling and that remains mechanistically particularly poorly understood. The MCTS1-DENR heterodimer is so far the only reported *trans*-acting factor that is specifically required in the re-initiation process, i.e. it does not appear to be essential for general translation as well—quite in contrast to several other proteins that have been linked to re-initiation in mammals, such as components of eIF3 (Bohlen et al, 2020a; Hronová et al, 2017; Shu et al, 2022) or eIF4 (Bohlen et al, 2020a). For these general eIFs, the analysis of specific re-initiation effects in vivo, for example, through loss-of-function experiments, is challenging, as they cannot easily be deconvoluted from abundant global translational alterations and their secondary effects. A defined in vitro assay that faithfully recapitulates and quantifies re-initiation rates gives the possibility of dissecting this process specifically, allowing for targeted, better interpretable experiments. Therefore, we view the development of

the in vitro assay presented in this study as an important methodological advance that will allow the characterisation of *cis*-acting mRNA/uORF sequence requirements and *trans*-acting protein machinery under defined conditions. Our careful benchmarking of the assay, including the use of different reporter constructs and variants and the demonstration of its dependence on MCTS1-DENR using knockout lysates and rescue with recombinant proteins, demonstrate the suitability of our system for detailed future mechanistic dissection. For example, a strategy to test the requirement for general eIFs in re-initiation could consist in producing HeLa in vitro translation lysates devoid of the protein of interest (e.g. through acute degron-mediated depletion, just before harvest of the HeLa cells), and to drive uORF translation on the *Klhdc8a* reporter through IRES sequences (e.g. the "factorless" Cricket Paralysis Virus IGR IRES (Pestova and Hellen, 2003)) that bypass the requirement for the particular eIF of interest. Thus, it should be possible to query the eIF requirement specifically in re-initiation. Other attractive future applications for our in vitro assay include compound screens (i.e. testing for specific inhibitors of the re-initiation process) and structural biology approaches (i.e. cryo-EM on uORF post-termination or other re-initiation-relevant complexes). The latter could also allow pinpointing where MCTS1-DENR acts in the overall re-initiation process, along the way from ribosome recycling to the actual assembly of the newly re-initiating ribosome on the downstream start codon, which remains an unresolved question. Finally, for future outlook on the in vitro assay, there may be interesting opportunities for more sophisticated modelling on the calculation of ribosome fluxes i.e., quantifying the likely paths of ribosomes along the reporter mRNAs by more refined approaches than the simple subtractive procedure used for now (Figs. 1J, 2F,H, 3B,D). In particular, it is quite possible that we are currently underestimating the true re-initiation efficiency (i.e. the calculated percentages represent lower limits), because the likely delay caused by translation of the uORF in the wild-type reporter may limit the maximally possible number of re-initiation events.

In addition to the methodological advancement, our study provides important new biological and mechanistic insights, notably regarding MCTS1-DENR and the paralogous MCTS2, as well as the role that eIF2D plays in gene expression regulation. In particular, our observations on eIF2D are surprising, as the cellular in vivo and in vitro evidence that we provide indicates that the main functions of this protein may not lie in uORF re-initiation after all. In retrospect, the case of eIF2D carrying re-initiation activity in mammals in vivo was always built on circumstantial evidence, as also noted in a recent review on this topic (Grove et al, 2024). This evidence was, mainly, (i) the clear homology of eIF2D to MCTS1-DENR, whose implication in the re-initiation process is undisputed; (ii) the in vitro ability shared by both factors to act on 40S subunits and deliver Met-tRNA$^{Met}_i$ to the empty P-site ((Skabkin et al, 2010); using an experimental model where 40S is in complex with HCV IRES-like mRNA—it should be noted that proof of equivalent structures on cellular mRNAs or in vivo is still lacking); (iii) an in vitro role of eIF2D for another viral element-dependent translational shuttling phenomenon that resembles re-initiation (Zinoviev et al, 2015); (iv) the shared in vitro activity to promote the release of deacylated tRNA and mRNA from recycled 40S subunits (Skabkin et al, 2010); (v) the high structural similarities of the proteins in reconstituted complexes together with the 40S ribosome (Ahmed et al, 2018; Lomakin et al, 2017; Weisser et al, 2017); and, finally, (vi) in vivo evidence in yeast that the orthologous proteins have shared functions which, however, do not lie in uORF re-initiation but

in post-termination tRNA removal and 40S ribosome recycling (Young et al, 2018)—events from which re-initiation can be envisaged to have subsequently evolved in higher eukaryotes. Last but not least, two recent studies have observed that eIF2D loss-of-function affected the expression of ATF4 in the integrated stress response in *Drosophila* (Vasudevan et al, 2020) and of reporter constructs that carried the *Atf4* 5′ UTR (or several other 5′ UTRs containing uORFs) in HeLa cells (Bohlen et al, 2020b). These effects were exacerbated by simultaneous DENR loss-of-function, which was interpreted as partially overlapping re-initiation activities of eIF2D and MCTS1-DENR. However, whether the eIF2D-dependent effects indeed relied on the mRNAs' uORFs, or whether they could be mechanistically distinct, was not specifically addressed. Notably, evidence has been accumulating that eIF2D plays roles separate from MCTS1-DENR (Alghoul et al, 2021; Rehfeld et al, 2023; Sonobe et al, 2021; Young et al, 2021), and the data presented in our study supports this idea as well.

Our data, based on the transcriptome-wide analysis of how RNA abundances and translation are altered upon eIF2D depletion, provides evidence that eIF2D is an important regulator of cellular gene expression, as judged from the massive reprogramming at the transcriptome level with hundreds of mRNAs up- and down-regulated and a cell cycle phenotype. However, minor changes occur at the translational level and, moreover, the affected mRNAs are not enriched for uORFs, which points to a main cellular role that does not involve re-initiation and therefore does not overlap with that of MCTS1-DENR. The specific analysis of *Atf4* mRNA in our datasets also supports this model, as we observe increased footprint abundance on the *Atf4* uORF1 stop codon in *Denr* shRNA-treated cells (in line with ribosomal stalling when deacylated tRNA removal is impaired (Bohlen et al, 2020b)), but not so in *Eif2d* shRNA-treated cells whose profiles are identical to control cells (Appendix Fig. S7). Still, since our in vitro experiments also reveal that high concentrations of eIF2D have the ability to compensate for MCTS1-DENR loss, it is quite likely that eIF2D has retained the capacity to act as a re-initiation factor. Whether this activity comes to bear under certain conditions or in specific cell types in vivo, will require careful further investigations. Probably the most revealing experiment to elucidate the biological function and mechanism by which eIF2D acts, and to distinguish direct from indirect targets in our profiling analyses, would be to identify the mRNAs and sequence elements that eIF2D physically interacts with in vivo, for example, through selective 40S footprinting (Bohlen et al, 2020a). Our efforts in this direction have, unfortunately, not yet been successful.

That MCTS2 can act as a bona fide, functionally active DENR interaction partner that is expressed to significant levels in the NIH/3T3 cells used for our study, is an important finding as it potentially explains some of the enigmatic published discrepancies between *Mcts1* vs. *Denr* knockout phenotypes. Thus, neuronal migration has been reported to be more strongly perturbed upon DENR loss-of-function than MCTS1 loss-of-function (Haas et al, 2016). Moreover, according to the International Mouse Phenotyping Consortium (IMPC; https://www.mousephenotype.org/), *Denr* knockout leads to preweaning lethality with complete penetrance, yet *Mcts1* knockouts are viable, with relatively mild reported phenotypes (mainly affecting eye morphology in early adult animals). The consortium did not phenotype knockouts of *Mcts2*, likely due to its pseudogene assignment. The high *Mcts2* RNA and footprint abundance observed in our study could be a particularity of NIH/3T3 cells, yet re-analysis of our

previous Ribo-seq data from mouse liver (Janich et al, 2015) and kidney (Castelo-Szekely et al, 2017) indicates that MCTS2 is co-expressed alongside MCTS1 in these organs as well, with different relative expression levels of the two paralogs (Appendix Fig. S8). Therefore, we propose that MCTS2 is a broadly expressed paralog. In the future, it will be particularly interesting to explore whether MCTS1-DENR and MCTS2-DENR heterodimers have redundant functions or whether the identity of the MCTS paralog has functional consequences for the activity of the heterodimer. Finally, while Bohlen et al recently reported that *Mcts2* expression is undetectable in various cells of the myeloid lineage (human whole blood cells, THP-1 cells, T-cell blasts) (Bohlen et al, 2023), our findings from mouse warrant a more thorough analysis across human tissue and cell types.

## Methods

### Reagents and tools table

| Reagent/resource | Reference or source | Identifier or catalogue number |
|---|---|---|
| **Experimental models** | | |
| HeLa cells | Teleman lab, Heidelberg | |
| HeLa Denr KO cells | Bohlen et al, 2020b | |
| HeLa Eif2d KO cells | This study | |
| HeLa double Denr Eif2d KO cells | This study | |
| NIH/3T3 | Schibler lab, Geneva | |
| HEK293FT | Schibler lab, Geneva | |
| bacterial strain BL-21 (DE-3) (pLysS) | Gruber Lab, Lausanne | |
| **Recombinant DNA** | | |
| pLV1 dual-luciferase reporter plasmid | Castelo-Szekely et al, 2019 | |
| Luciferase T7 control DNA plasmid | Promega | Cat. No. L4821 |
| Plasmids for MCTS1, DENR and eIF2D protein expression | Weisser et al, 2017 | |
| pCas9-2A-mCherry | Knuckles et al, 2017 | |
| pRRE200 | Flemr and Bühler, 2015 | |
| pBluescript II SK(+) | GenScript | |
| pX459 v2.0 | Addgene | #48139 |
| Non-functional lentiviral shRNA plasmids targeting Eif2d | Sigma | Cat. No. TRCN0000119842 |
| Lentiviral shRNA1 plasmids targeting *Eif2d* | Sigma | TRCN0000119844 |
| Lentiviral shRNA2 plasmids targeting *Eif2d* | Sigma | TRCN0000119845 |
| Scramble shRNA plasmids | Castelo-Szekely et al, 2019 | |
| Lentiviral shRNA2 plasmids targeting *Denr* | Castelo-Szekely et al, 2019 | |

| Reagent/resource | Reference or source | Identifier or catalogue number |
|---|---|---|
| pMD2.G | Addgene | #12259 |
| psPAX2 | Addgene | #12260 |
| GADD34Δ1-240 expression plasmid | Gurzeler et al, 2022 | |
| **Antibodies** | | |
| rabbit anti-KLHDC8A | Abcam | ab235419 |
| rabbit anti-VINCULIN | Abcam | ab129002 |
| mouse anti-U2AF65 | Sigma | U4758 |
| rabbit anti-EIF2D | Proteintech | 12840-1-AP |
| rabbit anti-DENR | Abcam | ab108221 |
| rabbit anti-phospho-eIF2alpha (Ser51) | Cell Signaling | #9721S |
| mouse anti-FLAG | Sigma | M2F3165 |
| rabbit anti-RPL23 | Abcam | ab112587 |
| mouse anti-beta-Tubulin | Sigma | T5201 |
| goat anti-guinea pig-HRP | Proteintech | SA00001-12 |
| anti-rabbit-HRP | Promega | W4011 |
| anti-mouse-HRP | Promega | S3721 |
| guinea pig anti-MCTS1 | Teleman lab, Heidelberg | Schleich et al, 2014 |
| **Oligonucleotides and other sequence-based reagents** | | |
| PCR primers and gRNAs | This study | Tables 1, 2, 3 & 4 |
| **Chemicals, Enzymes and other reagents** | | |
| Q5 site-directed mutagenesis kit | New England Biolabs | Cat. No. E0554S |
| BamHI | New England Biolabs | Cat. No. R0136S |
| XhoI | New England Biolabs | Cat. No. R0146S |
| EcoRV | New England Biolabs | Cat. No. R3195S |
| BsaI-HF v2 | New England Biolabs | Cat. No. R3733S |
| AatII | New England Biolabs | Cat. No. R0117S |
| SacI-HF | New England Biolabs | Cat. No. R3156S |
| SpeI-HF | New England Biolabs | Cat. No. R3133S |
| NdeI | New England Biolabs | Cat. No. R0111S |
| KpnI-HF | New England Biolabs | Cat. No. R3142S |
| DMEM, high glucose | Gibco™ | Cat. No. 41965062 |
| Penicillin-Streptomycin-Glutamine | Gibco™ | Cat. No. 10378016 |
| Foetal Bovine Serum | Gibco™ | Batch #2096980 |
| Lipofectamine™ 2000 | Invitrogen | Cat. No. 11668027 |
| Lipofectamine™ 3000 | Invitrogen | Cat. No. L3000008 |
| PhosSTOP™ | Roche | Cat. No. 4906845001 |

| Reagent/resource | Reference or source | Identifier or catalogue number |
|---|---|---|
| Puromycin | InvivoGen | Cat. No. 58-58-2 |
| 0.05% Trypsin-EDTA | Gibco™ | Cat. No. 25300054 |
| Dulbecco's phosphate-buffered saline | Gibco™ | Cat. No. 14190-169 |
| cOmplete™ EDTA-free Protease Inhibitor Cocktail | Roche | Cat. No. 4693132001 |
| IPTG | Invitrogen | Cat. No. 15529-019 |
| Imidazole | Sigma | Cat. No. I5513-5G |
| HisPur Ni-NTA resin slurry | Thermo Fisher Scientific | Cat. No. 88222 |
| Slide-A-Lyzer Dialysis Cassette (MWCO 10 K) | Thermo Fisher Scientific | Cat. No. 88400TS |
| MEGAscript® T7 Transcription Kit | Invitrogen | Cat. No. AM1334 |
| CleanCap Reagent AG (3' Ome) | TriLink | Cat. No. N-7413 |
| Yeast Inorganic Pyrophosphatase | New England BioLabs | Cat. No. M2403S |
| Turbo DNase | Invitrogen | Cat. No. AM2238 |
| Monarch RNA Cleanup Kit | New England BioLabs | Cat. No. T2040L |
| amino acids | Promega | Cat. No. L4461 |
| RNasin® Plus Ribonuclease Inhibitor | Promega | Cat. No. N2611 |
| Sodium creatine phosphate dibasic tetrahydrate | Sigma | Cat. No. 27920-1 G |
| Creatine Phosphokinase from rabbit muscle | Sigma | Cat. No. C3755-3.5KU |
| Dual-Glo® Luciferase Assay System | Promega | Cat. No. E2940 |
| passive lysis buffer | Promega | Cat. No. E2940 |
| Cycloheximide | Sigma | Cat. No. C7698-1G |
| Ambion™ RNase I | Invitrogen | AM2295 |
| MicroSpin S-400 columns | GE Healthcare | Cat. No. 27514001 |
| RiboCop rRNA Depletion Kit V1.2 | Lexogen | Cat. No. 037.24 |
| Click-iT® EdU Flow Cytometry Assay Kits | Invitrogen | Cat. No. C10424 |
| Propidium iodide | Invitrogen | Cat. No. P1304MP |
| dTAG-13 | Tocris | Cat. No. 6605 |
| Triton X-100 | Sigma | Cat. No. X-100-500ML |
| Pierce™ BCA Protein Assay kit | Thermo Fisher Scientific | Cat. No. 23225 |
| anti-FLAG magnetic beads | Millipore | Cat. No. M8823 |
| Trypsin/LysC mix | Promega | Cat. No. V5073 |
| cation exchange (SCX) plate (Oasis MCX) | Waters Corp., Milford, MA | 186001830BA |
| Acetonitrile LC-MS | Biosolve | Cat. No. 0001207802BS |
| Trifluoroacetic acid ULC-MS | Biosolve | Cat. No. 0020234139BS |
| Formic Acid, LC-MS Grade | Thermo Scientific | Cat. No. 85178 |
| ReproSil-Pur C18 custom-packed column | Dr. Maisch | |

| Reagent/resource | Reference or source | Identifier or catalogue number |
|---|---|---|
| **Software** | | |
| Cutadapt | Martin 2011 | |
| Bowtie2 | Langmead and Salzberg, 2012 | |
| STAR mapper version 2.5.3a | Dobin et al, 2013 | |
| StringTie | Pertea et al, 2015 | |
| edgeR | Robinson and Oshlack, 2010 | |
| Xcalibur | Thermo Scientific | |
| MaxQuant 1.6.14.0 | Tyanova et al, 2016 | |
| Perseus 1.6.15.0 | Tyanova et al, 2016 | |
| **Other** | | |
| Fusion Fx7 | Vilber | |
| ZentriMix 380 R centrifuge | Hettich AG | |
| Fragment Analyzer | Agilent | |
| Safire2™ plate reader | Tecan | |
| HiSeq2500 platform | Illumina | |
| Accuri C6 FACS | BD | |
| UltiMate™ 3000 RSLCnano System | Thermo Scientific | |
| Nanospray Flex™ Ion Sources | Thermo Scientific | |
| Orbitrap Exploris™ 480 Mass Spectrometer | Thermo Scientific | |
| Acclaim™ PepMap™ 100 C18 HPLC Columns | Thermo Scientific | |

## Cloning

To generate in vivo dual-luciferase reporter plasmids, we amplified the 5′ UTRs and first 5–10 (depending on construct) CDS codons (in frame with firefly luciferase CDS) of the selected transcripts by PCR from mouse cDNA or genomic DNA and cloned them into the BamHI restriction site of pLV1 dual-luciferase reporter plasmid with mutated firefly CDS start, essentially as previously described (Castelo-Szekely et al, 2019). Sequences of the primers used for the PCR are reported in Table 1. All constructs were validated by Sanger sequencing.

In vitro translation reporters were generated by PCR amplification of *Asb8* or *Klhdc8a* 5′ UTRs together with the six first CDS codons (in frame with the firefly luciferase CDS, lacking additional ATG start codon), from mouse cDNA, genomic DNA, or the corresponding pLV1 in vivo reporter constructs. The PCR products were then cloned into the BamHI restriction site of luciferase T7 control DNA plasmid (Cat. No. L4821, Promega). Modifications of uORFs or other mutations were performed using the Q5 site-directed mutagenesis kit (Cat. No. E0554S, New England Biolabs) and validated by Sanger sequencing. Sequences of primers for PCRs and mutagenesis are reported in Table 2.

Plasmids for MCTS1, DENR and eIF2D protein expression have been reported (Weisser et al, 2017), and cloning of human *Mcts2* cDNA for recombinant expression was analogous to that of *Mcts1*; briefly, a fragment encoding the first 154 amino acids of MCTS2 (i.e. the part carrying substitutions with regard to MCTS1) was PCR-amplified from HeLa genomic DNA with primers reported in Table 3 and cloned into the XhoI and EcoRV restriction sites of the pLIC C-terminally 6×His-TEV-tagged MCTS1-expression vector (Weisser et al, 2017). The correct sequence was validated by Sanger sequencing.

For endogenous tagging of DENR, a gRNA (GTCATCCACCTA-GAAGACACAGG) targeting a downstream sequence of the *Denr* stop codon was inserted into vectors pCas9-2A-mCherry (Knuckles et al, 2017) and pRRE200 (Flemr and Bühler, 2015) using oligos described in Table 4. About 5 µM of oligos were annealed in an annealing buffer (100 µM Tris-HCl pH 7.5, 500 mM NaCl and 10 mM EDTA) by cooling from 98 °C to 20 °C with a ramp of 0.02 °C per second. Annealed oligos were diluted tenfold, then ligated into pCas9-2A-mCherry BsaI restriction site or pRRE200 AatII and SacI restriction sites. Colony PCRs using the reverse gRNA oligo and a pCas9-2A-mCherry or pRRE200 forward primer was carried out to check for correct insertion of the annealed oligos and the cloned plasmids validated by Sanger sequencing.

A pBluescript II SK(+) vector was ordered as a synthetic clone to contain the knock-in cassette 3×FLAG-NeonGreen-dTAG. *Denr* 5′ and 3′ homology arms were amplified from mouse genomic DNA by PCR using primers reported in Table 4 and ligated in the pBluescript II SK(+) first digested with SpeI and BamHI for the 5′ homology arm, then with NdeI and KpnI for the 3′ homology arm. Positive clones were identified by colony PCR using PCR homology arm primers and by Sanger sequencing.

## Cell culture and generation of HeLa knockout cells

NIH/3T3, HEK293FT and HeLa cells were cultured under standard conditions (DMEM; 10% FCS, 1% penicillin/streptomycin, all from Invitrogen; 37 °C; 5% $CO_2$). The HeLa *Denr* knockout (KO) cells have been published (Bohlen et al, 2020b). For *Eif2d* KOs, wild-type HeLa cells were seeded in a six-well plate (200,000 cells per well) in DMEM with 10% FBS and 1% Pen/Strep. After 24 h, the cells were transfected with plasmid pX459 v2.0 into which a sgRNA sequence targeting *Eif2d* was cloned (sequence: CACCGCTGTGGACTG-GAAACACCCG) using Lipofectamine 2000 following the manufacturer's instructions. Twenty-four hours later, lipofectamine was removed, and cells were kept in DMEM for recovery for 2 days. On the third day, puromycin was added to a final concentration of 1.5 µg/ml. After another 3 days, puromycin was removed, and a fresh medium was added. After 2 days, cells were seeded into a 96-well plate at limiting dilution, so that only one cell per well was seeded. Wells were checked under the microscope for single-cell colonies. Upon reaching confluency, wells with only one colony were reseeded into six-well plates and kept for growing. Then immunoblots on these monoclonal lines were performed to identify the ones with loss of eIF2D. The knockout was verified at the genomic level via PCR using the following primers: 5′-GGGG AGGGCTCGGGTATGAC-3′ and 5′-AAGGCAGGGGCTGTCTC ATATC-3′ to amplify the genomic region. This revealed premature stop codons in all alleles. For generating *Eif2d* / *Denr* double knockout cells, the same procedure was performed to introduce the

**Table 1. Sequences of oligonucleotide primers used for the in vivo reporters.**

| Transcripts | Primer sequences for cloning in prLV1 |
|---|---|
| *Klhdc8a* | Forward: aaaggatccGTCTCCGACCCTGTAGACACTGCAG |
| | Reverse: tttggatccactagtATTGGGCACTTCCATGGCAGCCCGG |
| *Klhdc8a* start mut | Forward: GGCAGGGAGCCCGTGCAGCGCGgtaccGAGGCTGAGAGAGGGGACGCGCC |
| | Reverse: GGCGCGTCCCCTCTCTCAGCCTCggtacCGCGCTGCACGGGCTCCCTGCC |
| *Klhdc8a* stop mut | Forward: CGGCGCCGTGcacCCCGCGGGCC |
| | Reverse: GTCGGCGCGTCCCCTCTCTC |
| *Med23* | Forward: aaaggatccAGAGCAAGAGAGAGCGGCG |
| | Reverse: tttggatccactagtGCTCTGCAGTTGCGTCTCCATC |
| *Rps20* | Forward: aaaggatccGTTCCGGGTCACAAAGCACC |
| | Reverse: tttggatccactagtGGGCGTCTTTCCGGTATCTTTAAATG |
| *Cenpa* | Forward: aaaggatccATTGGCTTCAGACCTTTATTCTCATTGG |
| | Reverse: tttggatccactagtTGGGGTCTGCGGTTTGCGAC |
| *Ndc80* | Forward: aaaggatccTGACGTCAGCGCGGGCG |
| | Reverse: tttggatccactagtACCACAGGTGGAAACTGAACTGCG |

**Table 2. Sequences of oligonucleotide primers used for the in vitro reporters.**

| Transcripts | Primer sequences for cloning in pT7luc |
|---|---|
| *Klhdc8a* start mut | Forward: ggtGAAGACGCCAAAAACATAAAG |
| | Reverse: ttgaATCCTGACATTGGGCACT |
| *Klhdc8a* stop mut | Forward: CGGCGCCGTGcacCCCGCGGGCC |
| | Reverse: GTCGGCGCGTCCCCTCTCTC |
| *Klhdc8a* deletion 499 bp 5′ UTR | Forward: GGTGTCAGCGGGGCAGCAAGCTTGGGAGGCCTTA |
| | Reverse: TAAGGCCTCCCAAGCTTGCTGCCCCGCTGACACC |
| *Klhdc8a* penultimate codon GTC (WT & start mut) | Forward: GACCGGCGCCGTCTAGCCCGCGG |
| | Reverse: GGCGCG TCCCCTCTCTCAGC |
| *Klhdc8a* penultimate codon GTC (stop mut) | Forward: GACCGGCGCCGTCCACCCCGCGG |
| | Reverse: GGCGCGTCCCCTCTCTCAGCCTCC |
| *Klhdc8a* penultimate codon GCG (WT & start mut) | Forward: GACCGGCGCCGCGTAGCCCGCGG |
| | Reverse: GGCGCGTCCCCTCTCTCAGCC |
| *Klhdc8a* penultimate codon GCG (stop mut) | Forward: GACCGGCGCCGCGCACCCCGCGG |
| | Reverse: GGCGCGTCCCCTCTCTCAGCCTCC |
| *Asb8* | Forward: aaaggatccTTCTGCTTCCGGGTCACGCC |
| | Reverse: tttggatcctCCACATGCTGGAACTCATCAAGG |
| *Asb8* start mut | Forward: CGTTTGGAGCaccTGAACACACTCTGAGCCTTGATGAG |
| | Reverse: AACCGCGGGTCAGATCGG |

*Eif2d* knockout into the previously published *Denr*-knockout HeLa line (Bohlen et al, 2020b).

## Lentiviral transduction

Lentiviral shRNA plasmids targeting *Eif2d* were purchased from Sigma (Cat. No. TRCN0000119844 for shRNA1 and TRCN0000119845 for shRNA2). The *Denr* shRNA and the con1 shRNA have been previously described (see (Castelo-Szekely et al, 2019), where they are named "*Denr* shRNA2" and Scramble shRNA, respectively). The lentiviral shRNA plasmid for con2 shRNA was purchased from Sigma (Cat. No. TRCN0000119842) and was originally designed as an shRNA that can target *Eif2d*, yet as it shows no knockdown activity whatsoever (non-functional/nf shRNA) it is used as an additional control shRNA. Production of lentiviral particles in HEK293FT cells using envelope pMD2.G and packaging psPAX2 plasmids and viral transduction of NIH/3T3 cells were performed following published protocols (Castelo-Szekely et al, 2019; Salmon and Trono, 2007), with puromycin selection at 5 μg/ml for 4 days for shRNA-transduced cells.

## Western blot

Total protein extracts from NIH/3T3 and HeLa cells were prepared according to the NUN method (Lavery and Schibler, 1993). For analysis of phosphorylated proteins, PhosSTOP™ (Cat. No. 4906845001, Roche) was included in the NUN buffer. Antibodies used were rabbit anti-KLHDC8A 1:5000 (ab235419, Abcam), rabbit anti-VINCULIN 1:5000 (ab129002, Abcam) mouse anti-U2AF65 1:5000 (U4758, Sigma), rabbit anti-eIF2D 1:750 (12840-1-AP, Proteintech), rabbit anti-DENR 1:5000 (ab108221, Abcam; used for all immunoblots apart from Appendix Fig. S6B), rabbit anti-DENR 1:1000 (PA5-54507, Invitrogen; used for Appendix Fig. S6B), rabbit anti-phospho-eIF2alpha (Ser51) 1:1000 (#9721S, Cell Signaling), mouse anti-FLAG 1:10000 (M2F3165, Sigma), rabbit anti-RPL23 (ab112587, Abcam), mouse anti-beta-Tubulin 1:5000 (T5201, Sigma) and secondaries goat anti-guinea pig-HRP 1:10000 (SA00001-12, Proteintech), anti-rabbit-HRP (W4011, Promega) and anti-mouse-HRP

**Table 3.** Sequences of oligonucleotide primers used for cloning the pLIC C-terminally 6x-His-TEV-tagged *Mcts2* vector.

| Transcripts | Primer sequences for cloning in pLIC |
|---|---|
| *hsMcts2* | Forward: AAACTCGAGTTCTAGAAATAATTTTGTTTAACTTTAAGAAGGAGATATAGATCATGTTCAAGAAGTTTGATGAAAAGG |
| | Reverse: AAAGATATCTTCTGCAGACATCTTCATGAC |

**Table 4.** Sequences of oligonucleotides used for the cloning of *Denr* gRNA and homology arms into the pCas9-2A-mCherry, pRRE2OO and pBluescript II SK(-).

| Plasmids | Primers type | Oligo sequences |
|---|---|---|
| pCas9-2A-mCherry | Annealing oligos | Forward: caccgGTCATCCACCTAGAAGACAC |
| | | Reverse: aaacGTGTCTTCTAGGTGGATGACc |
| | Colony PCR | Forward: GCCTATTTCCCATGATTCCTTCA |
| pRRE2OO | annealing oligos | Forward: GTCGTCATCCACCTAGAAGACACAGGTGAACGT |
| | | Reverse: aaacGTGTCTTCTAGGTGGATGACc |
| | Colony PCR | Forward: CCACTTTGCCTTTCTCTCCACAG |
| pBluescript II SK(-) | To PCR 5′ homology arm | Forward: aaaactagtATTTAAAATGTAGATGAAATTTCAAAATGAAG |
| | | Reverse: aaaggatccCTTCTTCACTTCTCCAAGG |
| | To PCR 3′ homology arm | Forward: aaaattaatTGATCCTGAAGTCATGTGTTTCTAACTG |
| | | Reverse: aaaggtaccTGCACGAGGCCCTGGGTTC |
| | Q5 mutagenesis | Forward: GTTCTTTCACgTGTGTCTTCTAG |
| | | Reverse: TTCATGAATTGTGGCCAAAAC |

1:10000 (S3721, Promega). The guinea pig anti-MCTS1 antibodies (used at 1:500) have been described (Ahmed et al, 2018). Blots were visualised with the Fusion Fx7 system.

## Translation-competent extract preparation

The overall protocol for extract preparation was an adaptation of that published by Gurzeler et al, 2022 (Gurzeler et al, 2022). Briefly, confluent 15-cm dishes of HeLa cells were washed once with PBS at 37 °C, then trypsinised with 0.05% Trypsin-EDTA (Cat. No. 25300054, Gibco™). Harvested cells were collected by centrifugation at $500 \times g$ for 5 min at 4 °C. The cell pellet was washed twice with ice-cold PBS and centrifugated at $200 \times g$ for 5 min at 4 °C. The cell pellet was resuspended in 1:1 lysis buffer (10 mM HEPES pH 7.3, 10 mM potassium acetate, 500 μM MgCl$_2$, 5 mM DTT and 1× cOmplete™ EDTA-free Protease Inhibitor Cocktail (Cat. No. 4693132001, Roche)). Cells were lysed by dual centrifugation at 500 RPM for 4 min at −5 °C using a ZentriMix 380R centrifuge (Hettich AG) with a 3206 rotor and 3209 adaptors. The resulting lysate was then centrifuged in a table-top centrifuge at $13,000 \times g$ at 4 °C for 10 min, and the supernatant was aliquoted and snap-frozen in liquid nitrogen; storage at −80 °C for up to 6 months.

## Expression and purification of recombinant proteins

Recombinant eIF2D, MCTS1 and DENR protein purification has been described (Weisser et al, 2017). Recombinant MCTS2 was expressed and purified identically to MCTS1. Recombinant protein purity can be appreciated in Appendix Fig. S2D. The protein sequences of the recombinant proteins are given in Appendix Fig. S2E. GADD34Δ1-240 expression plasmid has been described (Gurzeler et al, 2022); briefly, for protein preparation, bacterial strain BL-21 (DE-3) (pLysS) was transformed with the plasmid, grown in 1000 ml Luria broth at 37 °C until the OD600 reached 0.4–0.6. After induction (1 mM IPTG), bacteria were cultured for another 4 h and cells were harvested. Expression of recombinant GADD34Δ1-240 was validated on Coomassie gel. The cell pellet was resuspended in 50 ml lysis/wash buffer (50 mM Tris-HCl pH 7.5, 300 mM NaCl, 5% glycerol, 25 mM imidazole), lysed by sonication with $240 \times 1$ s pulses at 40% amplitude on ice and centrifuged at $18,000 \times g$ for 20 min at 4 °C. The supernatant was mixed with 1 ml HisPur Ni-NTA washed resin slurry (Cat. No. 88222, Thermo Fisher Scientific) for at least 2 h at 4 °C under continuous rotation. Unbound proteins and background were removed by washing the resin 2× with 50 ml of lysis/wash buffer, then bound proteins were eluted 2× with 3 ml elution buffer (lysis/wash buffer supplemented with 400 mM imidazole). The proteins were dialysed in protein reconstitution buffer (30 mM NaCl, 5 mM HEPES pH 7.3) overnight using a Slide-A-Lyzer Dialysis Cassette (MWCO 10 K) (Cat. No. 88400TS, Thermo Fisher Scientific) at 4 °C under agitation. Protein concentration was quantified with a nanodrop device.

## In vitro transcription

Reporter DNA templates were generated by PCR amplification from the cloned reporters (forward primer: GCGCGTTGGCCGATTCAT-TAATGC; reverse primer: T$_{100}$GGGAGCTCGCCCCCTCGGAG). In vitro transcription was carried out for 2 h using the MEGAscript® T7 Transcription Kit protocol (AM1334, Invitrogen). Capping was carried out co-transcriptionally using the CleanCap Reagent AG (3′ Ome) (N-7413, TriLink) and the Yeast Inorganic Pyrophosphatase (Cat. No. M2403S, New England BioLabs) following the CleanCap protocol. The capped RNAs were treated for 15 min at 37 °C with 0.14

U/µl Turbo DNase (AM2238, Invitrogen), then purified using the Monarch RNA Cleanup Kit (T2040L, New England Biolabs). The quality and size of the produced RNAs were assessed using an Agilent Fragment Analyzer.

## In vitro translation assay

About 6.5 µl of translation-competent extracts were freshly supplemented with 0.4 mM amino acids (L4461, Promega), 15 mM HEPES pH 7.3, 6 mM Creatine Phosphate, 102 ng/ml Creatine kinase, 28 mM potassium acetate, 1 mM MgCl₂, 24 mM KCl, 1 U/µl RNasin® Plus Ribonuclease Inhibitor (N2611, Promega) and 2 fmol/µl of Renilla luciferase reporter in a final volume of 12.5 µl prior to each in vitro translation assay. Unless specified, 16 ng/µl of recombinant GADD34Δ1-240 was included in the supplemented extracts. About 30 fmol/µl of firefly luciferase reporter were used per reaction. The in vitro translation assays were carried out at 37 °C for 50 min. Luminescence activity was quantified using the Dual-Glo® Luciferase Assay System (E2940, Promega) and a Tecan Safire2™ plate reader.

## In vivo dual-luciferase assay

NIH/3T3 cells were first stably transduced with the lentiviral dual-luciferase reporters containing the 5′ UTRs to be assayed (upstream of firefly luciferase) and Renilla luciferase (internal normalisation), bidirectionally driven by the Pgk1 promoter. After establishing the reporter-expressing bulk cell population and several passages to ensure stable expression, cells were transduced with the shRNA lentiviruses and selected on puromycin for 4 days before lysis in 1× passive lysis buffer (E1941, Promega) and dual-luciferase readout using the Dual-Glo® Luciferase Assay System (E2940, Promega) and a Tecan Safire2™ plate reader.

## Ribosome profiling and library preparation

One 15-centimetre dish of confluent NIH/3T3 cells was used per replicate. Ribosome profiling and parallel RNA-seq were performed in triplicate for cells transduced with Denr shRNA, Eif2d shRNA1, Eif2d shRNA2, con1 shRNA and con2 shRNA, closely following our previously reported protocol (Castelo-Szekely et al, 2019) with minor modifications as described below. Cells were washed in cold PBS (without cycloheximide), harvested by scraping down in a small volume of PBS (without cycloheximide), pelleted by brief centrifugation, and then flash frozen. Cell lysates were prepared as described (with cycloheximide) (Castelo-Szekely et al, 2019). For the generation of ribosome-protected fragments, cell lysates were treated with 5 units Ambion™ RNase I (Cat No. AM2295, Invitrogen) per OD260, and monosomes were purified on MicroSpin S-400 columns (GE Healthcare). For both Ribo- and RNA-seq, 2 µg of RNA was used for rRNA depletion following the RiboCop rRNA Depletion Kit V1.2 (Cat. No. 037.24, Lexogen). Finally, libraries were amplified using 10-14 PCR cycles, and sequenced (HiSeq2500 platform).

## Sequencing pre-processing, alignment, quantification

Sequencing reads were processed as described previously (Arpat et al, 2020). Briefly, sequencing adaptors were trimmed using Cutadapt (Martin, 2011), and the reads were size-filtered by a custom Python

script (26–35 nt for RPF and 21–60 nt for RNA). The reads were then aligned subsequently to mouse rRNA, human rRNA, mouse tRNA and mouse cDNA from Ensembl mouse database release 100 using Bowtie2 (Langmead and Salzberg, 2012). In order to estimate expressed isoforms, the RNA-seq reads were mapped in parallel to the mouse genome using STAR mapper version 2.5.3a (Dobin et al, 2013) and processed with StringTie (Pertea et al, 2015) to measure FPKM for each transcript. A custom Python script classified the transcripts into single expressed isoforms or multiple transcript isoforms. The mapped reads were counted on 5′ UTR or CDS of the protein-coding genes using a custom Python script. The location of the putative A-site of RPFs were estimated as 5′ position +16 for reads shorter than or equal to 31 nt and +17 for reads longer than 31 nt. Transcripts for which there were less than 10 counts in total of all samples were filtered out of the further analysis. Read counts were normalised using the trimmed mean of M-values (TMM) from edgeR (Robinson and Oshlack, 2010), then transformed to transcripts per million (TPM) to normalise for transcript length. Translation efficiencies were calculated as the $\log_2$-transformed ratio of normalised CDS footprint counts to normalised CDS RNA counts and averaged over replicates. All analyses were carried out on R version 4.2.1.

## Differential translation efficiency analysis

Ribosome profiling sequencing data were analysed using the deltaTE method (Chothani et al, 2019) in order to assess significant changes in translation efficiency and RNA abundance. The significance threshold was set to FDR <0.1.

## uORF annotation and analyses

For uORF annotation, transcripts were used for which reads could be unambiguously assigned to the 5′ UTR, i.e. transcripts that are the only protein-coding isoform expressed (single expressed isoform) (N = 6841). In order to include as many transcripts as possible for the annotations, genes with multiple protein-coding isoforms were also considered if all expressed isoforms had the same CDS start, in which case the transcript with the longest 5′ UTR was selected (N = 2425). For the selected transcripts, uORFs were annotated and considered as translated with the following criteria: (i) they started with AUG, CUG, GUG or UUG, (ii) they had an in-frame stop codon within the 5′ UTR, and (iii) they had a coverage of at least 33%. When several potential uORFs were overlapping, the one with the highest coverage (read count/uORF length) was considered.

## Cell cycle monitoring

The cell proliferation assay was performed using the Click-iT® EdU Flow Cytometry Assay Kits (C10424, Invitrogen) following the vendor's protocol with some modifications. Briefly, NIH/3T3 cells transduced with con1 shRNA, con2 shRNA, Denr shRNA and Eif2d shRNA1 and shRNA2 were seeded into 6-cm plates under puromycin selection. After 4 days, the cells were treated for 1 h with 10 µM of EdU, washed with PBS, trypsinised and collected into FACS tubes. The cells were then centrifuged at 1200 × g for 5 min and resuspended in PBS. The cells were counted, and all conditions were adjusted to the same number of cells. After washing with 3 ml of 1% BSA in PBS, the cells were resuspended in 100 µl fixative buffer and incubated

overnight at 4 °C. Another wash with 1% BSA in PBS followed by a 15 min incubation in 100 µl of 1×-saponin-based permeabilisation and wash reagent. About 200 µl of Click-iT reaction cocktail were added to each tube, and the cells were incubated for 30 min at room temperature. After a wash in 3 ml of 1% BSA in PBS, the cells were resuspended in 100 µl of 1×-saponin-based permeabilisation and wash reagent complemented with 5 µl of 20 mg/ml RNaseA and 2 µl of 1 mg/ml propidium iodide and incubated for 30 min at room temperature. Fluorescence was measured using the Accuri C6 FACS machine.

## Generation of endogenously tagged cell lines

C-terminal tagging of endogenous DENR with 3×FLAG-Neon-Green-dTAG in NIH/3T3 cells was carried out using CRISPR-Cas9 homology-directed repair. $10^5$ NIH/3T3 cells were seeded in a six-well plate 24 h prior to transfection. About 0.9 µg of the pBLU donor plasmid (containing 400 bp 5′ and 3′ homology arms directly upstream and downstream of *Denr* stop codon) as well as 0.5 µg of the pCas9-gRNA and 0.1 µg of the control pRRE reporter were transfected into NIH/3T3 using Lipofectamine™ 3000 (L3000008, Invitrogen). The medium was changed after 24 h and cells were FACS-sorted based on mCherry and EGPF expression in a 10 cm dish 48 h post-transfection. Cells were counted and diluted to have 1 cell per well of a 96-well plate. When cell clones became visible, they were preselected based on NeonGreen expression, then split into a 96-well plate for genotyping and a 12-well plate for maintenance. For genomic DNA extraction, cells were washed with PBS and then lysed in 20 µl of DNA lysis buffer (10 mM Tris-HCl pH 8, 0.5 mM EDTA, 0.5% Triton X-100, 0.5 mg/ml proteinase K). After incubation for 1 h at 55 °C, cooling on ice and further incubation for 10 min at 95 °C to inactivate proteinase K, this prep was used to check the insertion of the cassette into the *Denr* locus by PCR using primers upstream of the 5′ and downstream of the 3′ homology arms.

## Co-immunoprecipitation

Triplicate 15 cm plates of 80% confluent NIH/3T3 DENR-3×FLAG-NeonGreen-dTAG cells were treated with 500 nM dTAG-13 or control-treated for 5 h prior to harvesting. After washing with PBS, cells were scraped in 1 ml PBS (4 °C). Cells were collected by centrifugation at 500×g for 5 min at 4 °C and lysed in 400 µl IP buffer (50 mM Tris-HCl pH 7.5, 120 mM NaCl, 1% NP-40, 0.5 mM DTT, 10 mM MgCl₂, 1× PhosSTOP™ (4906837001, Roche), 1× cOmplete™ EDTA-free Protease Inhibitor Cocktail (Cat. No. 4693132001, Roche), 0.04 U/µl RNasin® Plus Ribonuclease Inhibitor (CN2611, Promega)) for 30 min on ice with mixing by pipetting every 10 min. Lysed cells were centrifuged for 7 min at 5000 × g at 4 °C, and the supernatant was collected. Proteins were quantified using Pierce™ BCA Protein Assay kit (Cat. No. 23225, Thermo Fisher Scientific) and 500 µl of 2 mg/ml proteins were incubated with washed anti-FLAG magnetic beads (Cat. No. M8823, Millipore) at 4 °C for 3 h on a rotating wheel. The flow-through was kept for Western Blot, and beads were washed twice in wash buffer (20 mM Tris-HCl pH 7.5, 100 mM NaCl, 0.5 mM DTT, 10 mM MgCl₂, 1× cOmplete™ EDTA-free Protease Inhibitor Cocktail (Cat. No. 4693132001, Roche), 0.04 U/µl RNasin® Plus

Ribonuclease Inhibitor (N2611, Promega)) with change of tubes between the first and the second wash. Proteins were eluted in 80 µl elution buffer (NH₄OH pH 11–12) and incubated for 15 min at 4 °C. The elution step was repeated twice and eluates were pooled, frozen and dried in a SpeedVac system.

## Mass-spectrometry and data analysis

Samples were digested following a modified version of the iST method (named miST method). Briefly, dried material was redissolved in 50 µl miST lysis buffer (1% Sodium deoxycholate, 100 mM Tris pH 8.6, 10 mM DTT), heated 5 min at 95 °C and diluted 1:1 (v:v) with water. Reduced disulfides were alkylated by adding 0.25 vol. of 160 mM chloroacetamide (32 mM final) and incubating for 45 min at RT in the dark. Samples were adjusted to 3 mM EDTA and digested with 1.0 µg Trypsin/LysC mix (Promega #V5073) for 1 h at 37 °C, followed by a second 1 h digestion with an additional 0.5 µg of proteases. To remove sodium deoxycholate, two sample volumes of isopropanol containing 1% TFA were added to the digests, and the samples were desalted on a strong cation exchange (SCX) plate (Oasis MCX; Waters Corp., Milford, MA) by centrifugation. After washing with isopropanol/1% TFA, peptides were eluted in 200 µl of 80% MeCN, 19% water, 1% (v/v) ammonia, and dried by centrifugal evaporation. Tryptic peptide mixtures were injected on an Ultimate RSLC 3000 nanoHPLC system interfaced via a nanospray Flex source to a high-resolution Orbitrap Exploris 480 mass spectrometer (Thermo Fisher, Bremen, Germany). Peptides were loaded onto a trapping microcolumn Acclaim PepMap100 C18 (20 mm × 100 µm ID, 5 µm, Dionex) before separation on a C18 custom-packed column (75 µm ID × 45 cm, 1.8-µm particles, Reprosil-Pur, Dr. Maisch), using a gradient from 4 to 90% acetonitrile in 0.1% formic acid for peptide separation (total time: 140 min). Full MS survey scans were performed at 120,000 resolution. A data-dependent acquisition method controlled by Xcalibur software (Thermo Fisher Scientific) was used that optimised the number of precursors selected ("top speed") of charge 2+ to 5+ while maintaining a fixed scan cycle of 2 s. Peptides were fragmented by higher energy collision dissociation (HCD) with a normalised energy of 30% at 15,000 resolution. The window for precursor isolation was of 1.6 *m/z* units around the precursor and selected fragments were excluded for 60 s from further analysis. Data files were analysed with MaxQuant 1.6.14.0 incorporating the Andromeda search engine. Cysteine carbamido-methylation was selected as a fixed modification, while methionine oxidation and protein N-terminal acetylation were specified as variable modifications. The sequence databases used for searching were the mouse *(Mus musculus)* reference proteome based on the UniProt database (www.uniprot.org, version of April 6, 2021, containing 55,341 sequences RefProt _Mus_musculus_20210604.-fasta) supplemented with the sequences of common contaminants, and a database of mammalian immunoglobulin sequences (https://www.imgt.org version of 20th Nov. 2018, 2243 entries). Mass tolerance was 4.5 ppm on precursors (after recalibration) and 20 ppm on MS/MS fragments. Both peptide and protein identifications were filtered at 1% FDR relative to hits against a decoy database built by reversing protein sequences. All subsequent analyses were done with the Perseus software package (version 1.6.15.0). Contaminant proteins were removed, and intensity iBAQ values

were log$_2$-transformed. After assignment to groups, only proteins quantified in at least three samples of one group were kept. After missing values imputation (based on normal distribution using Perseus default parameters), *t*-tests were carried out among all conditions, with permutation-based FDR correction for multiple testing (Q-value threshold <0.05). Log$_2$FC of non-treated Denr-3×FLAG-NeonGreen-dTAG vs. dTAG treated Denr-3×FLAG-NeonGreen-dTAG was calculated and the *p* value was corrected for multiple testing by false discovery rate (significance threshold was set to FDR <0.05).

## Data availability

Sequencing data have been deposited at GEO (GSE263991; https://www.ncbi.nlm.nih.gov/geo/query/acc.cgi?acc=GSE263991) and mass-spectrometry data at PRIDE (PXD051482; https://proteomecentral.proteomexchange.org/cgi/GetDataset?ID=PXD051482). All computational scripts are available at https://zenodo.org/records/14235820 and https://github.com/gatfieldlab/Re-initiation_project.

The source data of this paper are collected in the following database record: biostudies:S-SCDT-10_1038-S44318-024-00347-3.

## Peer review information

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

## Acknowledgements

We thank the Lausanne Genomic Technologies Facility for library sequencing and the Lausanne Protein Analysis Facility for mass-spectrometry-based proteomics. Evan Karousis for help with the initial setup of the in vitro translation assay and for the GADD34Δ1-240-expression plasmid, Marc Bühler and Fabio Mohn for advice on endogenous tagging, and Michael Taschner for his help with preparation of recombinant GADD34Δ1-240. We thank members of the Gatfield lab for their critical comments on the manuscript. This work was supported by Swiss National Science Foundation (SNSF) NCCR RNA & Disease Phase III funding, grant number 205601 (to D.G. and N.B.) and by SNSF Project funding, grant number 212423 (to D.G.).

## Author contributions

**Romane Meurs**: Conceptualisation; Software; Formal analysis; Investigation; Visualisation; Methodology; Writing—original draft; Writing—review and editing. **Mara De Matos**: Investigation; Methodology. **Adrian Bothe**: Resources; Visualisation; Methodology; Writing—review and editing. **Nicolas Guex**: Formal analysis; Visualisation. **Tobias Weber**: Resources; Methodology. **Aurelio A Teleman**: Resources; Supervision; Writing—review and editing. **Nenad Ban**: Resources; Supervision; Methodology; Writing—review and editing. **David Gatfield**: Conceptualisation; Supervision; Funding acquisition; Visualisation; Writing—original draft; Project administration; Writing—review and editing.

Source data underlying figure panels in this paper may have individual authorship assigned. Where available, figure panel/source data authorship is listed in the following database record: biostudies:S-SCDT-10_1038-S44318-024-00347-3.

## Disclosure and competing interests statement

The authors declare no competing interests.

