## [Peer Review File · The EMBO Journal]

MCTS2 and distinct eIF2D roles in uORF-dependent translation regulation revealed by in vitro re-initiation assays

Romane Meurs, Mara De Matos, Adrian Bothe, Nicolas Guex, Tobias Weber, Aurelio Teleman, Nenad Ban, and David Gatfield

Corresponding author(s): David Gatfield (david.gatfield@unil.ch)

Review Timeline:

Transfer from Review Commons:	3rd Oct 24
Editorial Decision:	28th Oct 24
Revision Received:	28th Nov 24
Accepted:	10th Dec 24

Editor: Cornelius Schneider

Transaction Report: This manuscript was transferred to The EMBO JOURNAL following peer review at Review Commons.

Review #1

1. Evidence, reproducibility and clarity:

Evidence, reproducibility and clarity (Required)

****Summary:****

The manuscript by Meurs et al describes the development of an in vitro translation system for the study of re-initiation events on uORFs in human RNAs, which is used to characterise initial findings from omics-based analyses on the dependence of different uORFs on various co-factors including MCTS1-DENR, eIF2D and a newly characterised MCTS2-DENR complex in detail.

****Major comment:****

1. On lines 405/406 and relating to data shown in figure 4J, the authors emphasize that "high" concentrations of eIF2D could achieve the "partial compensation" for MCTS1-DENR loss the authors observed. During reading I interpreted the "high" as indicating that eIF2D concentrations were higher than for MCTS1-DENR, but the legend to figure 4J states that the concentrations for both were 0.5 μ M. I think the language used here is biasing against the idea that eIF2D can be as efficient as MCTS1-DENR, when both concentrations are actually identical.

Related, the statement that eIF2D could only "partially" rescue MCTS1-DENR loss appears to rest on the comparison between the "WT" reporter with and without supplementation in figure 4J, which is slightly lower for eIF2D supplementation than for MCTS1-DENR supplementation. As far as I can see this particular comparison is not supported by any statistical analysis, and I doubt that the small difference would be significant.

On both counts, I am not sure that the statement that eIF2D can only partially rescue the effect of combined eIF2D and MCTS1-DENR depletion is robust - based on the data shown I think it is possible that eIF2D rescues re-initiation as efficiently as MCTS1-DENR when this complex is absent. It would be useful to adjust the conclusion here (or clarify the explanations if my interpretation of the results is wrong).

I note that this is different from the conclusion that eIF2D does not rescue defects when MCTS1-DENR is present, which is the main conclusion of the paper and is well supported.

****Minor Comments:****

2. In the sentence on lines 105-109, the authors note that transcripts with significant differences in TE upon DENR depletion had longer 5'-UTRs than the genome-wide average, which is consistent with the enrichment for uORF containing RNAs in this sample which have longer 5'-UTRs. Is there also any significant difference in the UTR lengths between DENR-regulated and non-DENR regulated uORF containing UTRs? Discussing this could be informative regarding the question why some uORF-containing transcripts are DENR dependent and some are not...

3. The beginning of the results section discusses translational efficiency on the main ORF as a

result of MCTS1-DENR depletion. Is it possible to say from the data whether translational efficiency/ footprint density on the uORFs themselves also changes? This could inform our understanding of what these factors actually do during re-initiation.

4. In figure 2B, the supernatant fraction (2) seems enriched for eIF2D. Do the authors know what causes this effect? I wondered whether this is related somehow to the fact that the general control pathway has been activated (as the eIF2 phosphorylation evidences), but struggle to come up with a mechanism that could increase protein amounts on the relatively short timescales of the preparation of the cell lysate.

5. In figure 2D, firefly luciferase activity in the replicates seems highly variable (over more than one order of magnitude, much more so than the Renilla luciferase used for normalisation). Why is this - did these samples come from different conditions? Within-sample variability in all subsequent analyses seems much less variable.

6. On line 152, it would be useful to briefly mention what GADD34 actually is

7. On line 64, the authors mention conservation of MCTS1-DENR in "Drosophila, mouse and human". On line 66, yeast orthologues are mentioned - it would slightly improve reading flow if yeast was also listed in the first sentence.

2. Significance:

Significance (Required)

In my view this is an interesting study that enhances our understanding of re-initiation on upstream ORFs, ubiquitous and important regulatory elements in eukaryotic mRNAs. The addition of in vitro translation-based assays to the available toolset (which is not commonly used to study uORF re-initiation) is a substantial contribution, and in this manuscript is combined with omics-based in vivo analyses to advance our understanding of specific re-initiation factors. The confirmation that MCTS1-DENR is a specific re-initiation factor on a subset of human RNAs, that eIF2D can support re-initiation but appears to have roles outside of this process in vivo, and that MCTS2 is a functional paralogue of MCTS1 are novel findings of interest to a wide scientific community.

3. How much time do you estimate the authors will need to complete the suggested revisions:

Estimated time to Complete Revisions (Required)

(Decision Recommendation)

Less than 1 month

4. Review Commons values the work of reviewers and encourages them to get credit for their work. Select 'Yes' below to register your reviewing activity at Web of Science Reviewer Recognition Service (formerly Publons); note that the content of your review will not be visible on Web of Science.

Yes

Review #2

1. Evidence, reproducibility and clarity:

Evidence, reproducibility and clarity (Required)

Summary

Translational reinitiation, a process of ribosome restart scanning of mRNA segments after termination of short ORF, has been broadly recognized in many transcripts. In addition to regular translation initiation factors, protein players specifically contributing to the reinitiation have been proposed: MCTS1-DENR complex and eIF2D. However, the quantitative assessment of these factors to reinitiation remained elusive.

The authors addressed MCTS1-DENR-mediated translational reinitiation by an elegant in vitro translation system in humans. This system faithfully recapitulated the uORF-dependent impacts of these factors on cellular translation, observed by ribosome profiling. Importantly, the authors quantitatively dissect the ribosome flux for leaky scanning to bypass uORF, reinitiation from uORF, and ribosome recycling after uORF translation. Strikingly, MCTS1-DENR contributes to the reinitiation process in a significant fraction, whereas eIF2D has marginal effects on that. The author also found that MCTS2, a paralog of MCTS1, has a similar function to MCTS1 for reinitiation.

Major comments

This manuscript is well written. The data strongly support the conclusion. This reviewer does not have a major comment for this manuscript. This reviewer has only several minor comments as described below.

Minor comments

1. In line 102, "non-functional (nf) Eif2d shRNA" sounds a bit confusing. Although this reviewer guessed that "non-functional" shRNA may be designed by shuffled sequence or so, the another wording may be easy for readers.
2. For Figure 1C, the caption for "Denr shRNA" looks a bit confusing. It should be "transcripts with low TE by Denr shRNA" or so.
3. In Supplementary Figure S3A, the authors treated cell lysate with tunicamycin and recapitulated UPR/ISR mediated by PERK on ER. This sounds interesting since even in the lysate, ER may be intact for UPR/ISR. The author should consider commenting on this in the text.
4. For Figure 3 and Supplementary Figure S3, the authors used the same jargon of "reinitiation" for the effects from both MCTS1-DENR and phosphorylated eIF2alpha. However, the points to which these factors contributed, should be different; MCTS1-DENR may function the departure of 40S

from termination codons (or blocking 60S rejoining), whereas phosphorylated eIF2alpha blocks the recruitment of initiator tRNA on 40S. More concrete wording in the paragraph may be helpful for the understanding of readers.

5. Related to the point above, the author may consider explaining why Atf4 and Atf5 are more sensitive to phosphorylated eIF2alpha than Asb8 and Klhdc8a.

6. For Figure 5H, sole DENR could partially rescue reinitiation deficiency in DENR KO lysate which also lost MCTS1. Although this reviewer guessed that this originated from the residual amount of MCTS1 in the lysate, the author should consider adding an explanation for this.

7. The author should add a Western blot for MCTS2 in DENR KO cells to test whether DENR loss also affects the stability of MCTS2 as seen in MCTS1.

2. Significance:

Significance (Required)

Translational reinitiation, a process of ribosome restart scanning of mRNA segments after termination of short ORF, has been broadly recognized in many transcripts. In addition to regular translation initiation factors, protein players specifically contributing to the reinitiation have been proposed: MCTS1-DENR complex and eIF2D. However, the quantitative assessment of these factors to reinitiation remained elusive.

The authors addressed MCTS1-DENR-mediated translational reinitiation by an elegant in vitro translation system in humans. This system faithfully recapitulated the uORF-dependent impacts of these factors on cellular translation, observed by ribosome profiling. Importantly, the authors quantitatively dissect the ribosome flux for leaky scanning to bypass uORF, reinitiation from uORF, and ribosome recycling after uORF translation. Strikingly, MCTS1-DENR contributes to the reinitiation process in a significant fraction, whereas eIF2D has marginal effects on that. The author also found that MCTS2, a paralog of MCTS1, has a similar function to MCTS1 for reinitiation. This work provides important findings for broad readers not only in the RNA/translation field but also in immunity and neurological disorders in which MCTS1-DENR is involved.

3. How much time do you estimate the authors will need to complete the suggested revisions:

Estimated time to Complete Revisions (Required)

(Decision Recommendation)

Less than 1 month

4. Review Commons values the work of reviewers and encourages them to get credit for their work. Select 'Yes' below to register your reviewing activity at Web of Science Reviewer Recognition Service (formerly Publons); note that the content of your review will not be visible on Web of Science.

Yes

Review #3

1. Evidence, reproducibility and clarity:

Evidence, reproducibility and clarity (Required)

Meurs et al. provide evidence for the role of DENR and MCTS1, as well as a the MCTS2 paralog, in translational control regulated by re-initiation after uORFs. These studies are based off of reporter assays in mammalian in vitro translation lysates prepared from WT or depleted cells, as well as ribosome profiling and RNA-seq. The experiments are well controlled, appropriate for the questions being asked, and incorporate rigorous controls and methods. The manuscript does provide new insight into DENR, MCTS1, MCTS2, and eIF2D that is informative for the field. However, my interpretation of the tone and model from the authors is that DENR and MCT1/2 are directly acting in re-initiation (i.e., at the re-initiated start codon)-however, their data can also be explained by the counter model in the field that these factors are in fact ribosome recycling factors. This does not lower the validity of these new data, but I feel they should be presented with both models in mind and equally plausible. From what I can tell, the authors have not tested the two models; I don't think they need to at the moment but they should present both models equally.

****My comments are:****

The authors reference published structural work supporting that these alternative initiation factors in fact do directly act in re-initiation (at the re-initiated start codon); however, these studies used a truncated/mutated viral IRES and the overall conclusion has not been shown directly on any mammalian mRNA. I apologize if I missed the latter, but if so, please directly state this and the evidence.

Figure 1 layout is not too intuitive as it goes in different directions; additionally, 1J is referenced in the text before 1H and 1I.

In Figure 1F, the increase in RPFs in the Denr KD is highly consistent with a defect in ribosome recycling, no re-initiation per se. This appears to be glanced over and not tied into any of the downstream hypotheses. (see below regarding comment about ribosome recycling from PMID 30146315 and 34016977).

Just to avoid potential issues as some Denr, MCTS1 and eIF2D studies as done in animal models, the use of "in vivo" in this paper should be more accurately described as "in 3T3/HeLa cells"

In Figure 2D, the use of two subtle gray tones that overlap makes the data incomprehensible. I assume the data on the left in each major group (RLuc and FFLuc) is +GADD34?

For line 239-240, the line "Taken together, these results suggest that eIF2 α levels are only

moderately limiting for re-initiation." should be more accurate. For example, "Taken together, these results suggest that phosphorylated eIF2 α levels are only moderately limiting for re-initiation on Klhdc8a and Asb8 mRNAs in vitro."

The line on page 10, "This measure is expected to increase when a re-initiation factor is depleted because ribosomes are selectively lost from CDS sequences but will still translate uORFs when re-initiation is inhibited [17]." Isn't this more consistent with role of MCTS1 and DENR in 40S recycling at stop codons (not re-initiation per se) as shown by the Guydosh Lab? They have also shown that MCTS1 and DENR play a bigger role than eIF2D in this role, at least in yeast-PMID 30146315 and 34016977. In general, I feel this point should be brought up in the start of the manuscript. While I don't disagree with the included data, but the impact on recycling at the identified uORFs (not direct re-initiation at the main/primary ORF start codon) can also easily explain the observed effects with DENR and MCTS1.

The authors should provide evidence that their recombinant proteins used in this study are highly pure.

Cricket Paralysis Virus IRES should be Cricket Paralysis Virus IGR IRES. CrPV has two reported IRESs. The IGR is the "factorless" one.

In the Discussion, the critique of the eIF2D and related data in the field has been recently described elsewhere too-PMID 38433101.

2. Significance:

Significance (Required)

The manuscript does provide new insight into DENR, MCTS1, MCTS2, and eIF2D that is informative for the field. However, my interpretation of the tone and model from the authors is that DENR and MCT1/2 are directly acting in re-initiation (i.e., at the re-initiated start codon)-however, their data can also be explained by the counter model in the field that these factors are in fact ribosome recycling factors. This does not lower the validity of these new data, but I feel they should be presented with both models in mind and equally plausible. From what I can tell, the authors have not tested the two models; I don't think they need to at the moment but they should present both models equally.

3. How much time do you estimate the authors will need to complete the suggested revisions:

Estimated time to Complete Revisions (Required)

(Decision Recommendation)

Less than 1 month

4. Review Commons values the work of reviewers and encourages them to get credit for their work. Select 'Yes' below to register your reviewing activity at Web of Science Reviewer Recognition Service (formerly Publons); note that the content of your review will not be visible on Web of Science.

No

Review #4

1. Evidence, reproducibility and clarity:

Evidence, reproducibility and clarity (Required)

****Summary:**** In this manuscript, Meurs and colleagues describe a multifaceted analysis of the roles of eIF2D, DENR, MCTS1 and MCTS2 in translation reinitiation. Translation reinitiation occurs primarily after ribosomes translate short upstream Open Reading Frames (uORFs). Previous work has established that DENR/MCTS1 contributes to reinitiation after translation of many uORFs, to varying extents. It has been suggested that eIF2D is also a reinitiation factor due to structural similarity to the MCTS1/DENR complex. Meurs et al used a clever combination of in vivo and in vitro reporter assays and ribosome profiling to evaluate the roles of these factors on reinitiation after translation of several uORFs. The manuscript is well written, addresses an important knowledge gap in translation, and uses a satisfyingly elegant set of experiments to quantitate leaky scanning and translation reinitiation. The main novel findings are that the reinitiation function of DENR/MCTS1 varies across uORFs, that eIF2D can partially substitute for DENR reinitiation activity in vitro, and that MCTS2 is expressed and functions in reinitiation. MCTS2 is an intronless gene that was created by retrotransposition, and was previously reported to be a non-functional pseudogene. Overall, there is much to appreciate, with only a few areas that could be addressed. Numbered comments follow:

1. Figure 2J - The authors say that lengthening the ASB8 uORF appears to reduce DENR dependence for reinitiation. Is this difference statistically significant?
2. As the authors acknowledge, the lack of apparent translation efficiency changes in the eIF2D KD / deletion may result from its activity being redundant with DENR-MCTS1. Do mRNA levels or translation of MCTS1, MCTS2, or DENR change with eIF2D deletion or depletion?
3. It's a little disappointing that the authors can't clearly determine whether eIF2D functions in reinitiation, in part because eIF2D deletion or depletion lead to a cell cycle defect. They suggest future work with an auxin degree version of eIF2D and depletion in the cell. Could they also use immunodepletion in extracts to more directly evaluate eIF2D in this study?
4. The graphed reporter results include shaded areas representing uORF repression, leaky scanning, reinitiation, and DENR dependence. While this is very concisely presented, it is also difficult to read the graphs. Could the authors provide a mini bar graph (perhaps log scale) to show this more clearly?
5. On line 233, the authors describe the Klhdc8a uORF-CDS distance as 73 nucleotides, but it looked shorter to this reviewer (60 nt). Could you please double check that?
6. It is great to see the authors checked the eIF2-alpha phosphorylation state in their in vitro extract preparations. However, in some ways this opens up a "can of worms" regarding in vitro translation

systems in general. One can't check all stress-related modifications. For example, are ribosomes ubiquitinated due to the RQC pathway activation in extracts? Are any other initiation factors post-transcriptionally modified? This is not necessarily a criticism of this paper or its use of in vitro translation, but it is more a comment on in vitro translation systems in general. The authors may (or may not) want to add this to the discussion.

2. Significance:

Significance (Required)

General Assessment: Overall, this manuscript reports very high quality work. Experimental design was excellent, data analysis was well done, the manuscript is well-written and the figures are very clear for the most part.

Advance: The main advance of the paper is that it shows that eIF2D can replace the function of MCTS1-DENR and that MCTS2 is a functional gene. The in vitro assay system is a modest technological advance (in vitro translation systems are commonly used), but it is highly appreciated that the authors checked eIF2-alpha phosphorylation in the extract.

Audience: The audience for this paper would be primarily a gene expression / mRNA translation audience.

3. How much time do you estimate the authors will need to complete the suggested revisions:

Estimated time to Complete Revisions (Required)

(Decision Recommendation)

Less than 1 month

4. Review Commons values the work of reviewers and encourages them to get credit for their work. Select 'Yes' below to register your reviewing activity at Web of Science Reviewer Recognition Service (formerly Publons); note that the content of your review will not be visible on Web of Science.

Yes

Full Revision

Manuscript number: RC-2024-02551

Corresponding author(s): David Gatfield

1. General Statements

We are grateful to the four Reviewers for their detailed assessment of our manuscript and are delighted about their constructive and very positive evaluations, highlighting the study's novelty, rigor and mechanistic contributions to the field.

We have addressed all remaining issues and added further analyses; we have also extensively rewritten parts of the manuscript, which has greatly improved as a consequence. We hope that the Reviewers will find the outcome satisfactory as well.

Reviewer #1

Evidence, reproducibility and clarity (Required):

Summary:

The manuscript by Meurs et al describes the development of an in vitro translation system for the study of re-initiation events on uORFs in human RNAs, which is used to characterise initial findings from omics-based analyses on the dependence of different uORFs on various co-factors including MCTS1-DENR, eIF2D and a newly characterised MCTS2-DENR complex in detail.

Major comment:

1) On lines 405/406 and relating to data shown in figure 4J, the authors emphasize that "high" concentrations of eIF2D could achieve the "partial compensation" for MCTS1-DENR loss the authors observed. During reading I interpreted the "high" as indicating that eIF2D concentrations were higher than for MCTS1-DENR, but the legend to figure 4J states that the concentrations for both were 0.5 μ M. I think the language used here is biasing against the idea that eIF2D can be as efficient as MCTS1-DENR, when both concentrations are actually identical.

Related, the statement that eIF2D could only "partially" rescue MCTS1-DENR loss appears to rest on the comparison between the "WT" reporter with and without supplementation in figure 4J, which is slightly lower for eIF2D supplementation than for MCTS1-DENR supplementation. As far as I can see this particular comparison is not supported by any statistical analysis, and I doubt that the small difference would be significant.

On both counts, I am not sure that the statement that eIF2D can only partially rescue the effect of combined eIF2D and MCTS1-DENR depletion is robust - based on the data shown I think it is possible that eIF2D rescues re-initiation as efficiently as MCTS1-DENR when this complex

is absent. It would be useful to adjust the conclusion here (or clarify the explanations if my interpretation of the results is wrong).

I note that this is different from the conclusion that eIF2D does not rescue defects when MCTS1-DENR is present, which is the main conclusion of the paper and is well supported.

Author response: The Reviewer's point is well-taken and we have rephrased this part of the results section to avoid misunderstanding or biased/wrong interpretation.

Briefly, our finding in Figure 4J is that the double- (*Denr* and *Eif2d*)-KO lysates have low residual re-initiation activity (left panel). Adding recombinant MCTS1-DENR at 0.5 μ M rescues most re-initiation activity in the double-KO lysates (right panel). However, supplementing with eIF2D at the same concentration, i.e. 0.5 μ M, would appear to rescue re-initiation activity to a lesser extent (middle panel; thus, the WT-uORF reporter still shows a statistically significant difference in its signal when comparing WT lysate +recombinant eIF2D to double-KO lysate +recombinant eIF2D). For these reasons we had originally used the term "partial rescue".

We have now specifically used 2-way ANOVA to test if there was a significant interaction when comparing double KO lysate rescued with recombinant eIF2D and double KO lysate rescued with recombinant MCTS1-DENR. We did not find a significant interaction. The Reviewer is therefore right, we cannot claim that the rescue is only partial for eIF2D.

Finally, we had used the somewhat misleading (and subjective) term "high concentration" because 0.5 μ M is likely a relatively high concentration compared to that of the endogenous eIF2D protein in wild-type cells or lysates (assuming that a typical, low/medium-abundance protein may rather have a concentration around 0.1 μ M or lower). However, we do not have quantitative data on the actual endogenous eIF2D protein concentration. Therefore, our statement of "high levels" was not a good choice and we have now changed the text accordingly. Just as a final piece of information: in assay setup experiments not included in the manuscript, we have previously titrated recombinant eIF2D levels up to final 2 μ M, yet these higher concentrations also did not increase rescue (data not shown); thus, eIF2D is not rate-limiting in our assay.

We have now rephrased this whole paragraph of the results section as follows and hope that the Reviewer agrees with the new version (see lines 330-348 in manuscript, non-track-changed version):

"In *Eif2d* knockout extracts, MCTS1-DENR is abundant and may mask the re-initiation function of eIF2D. To address this possibility, we carried out two experiments. First, using re-initiation-deficient *Denr* KO extracts and recombinant eIF2D, we were indeed able to partially rescue re-initiation activity (**Figure 4I**), although not as efficiently as with recombinant MCTS1-DENR (compare with **Figure 2E**). Second, we used *Denr* + *Eif2d* HeLa double knockout cell lysates. *In vitro* translation reactions in these extracts showed reduced re-initiation activity (**Figure 4J**, left) that was very similar to *Denr* single knockouts (**Figure 2E**). Addition of recombinant MCTS1-DENR (0.5 μ M) showed the expected rescue (**Figure 4J**, right). Interestingly, also recombinant eIF2D (0.5 μ M) showed rescue of the double knockout (**Figure 4J**, middle). Although rescue with recombinant eIF2D appeared less potent than rescue with recombinant MCTS1-DENR at first sight – i.e., in the right panel to **Figure 4J**, there is no significant difference on WT uORF reporter between MCTS1-DENR-supplemented WT vs. MCTS1-

DENR-supplemented double KO lysates, whereas in the middle panel, double KO lysate supplemented with recombinant eIF2D still showed significantly lower signal than WT lysate supplemented with eIF2D – 2-way ANOVA did not uncover a significant interaction and we concluded that eIF2D can potentially rescue as potently as MCTS1-DENR. Taken together, these results indicate that eIF2D is likely not an essential component of the re-initiation machinery in cells *in vivo* and *in vitro* under wild-type conditions where MCTS1-DENR is abundantly present. However, when MCTS1-DENR is absent, providing eIF2D protein can compensate for MCTS1-DENR loss. Thus, it is a probable scenario that eIF2D has in principle retained shared activity with MCTS1-DENR, but we deem it likely that its main cellular functions under physiological conditions lie outside of uORF re-initiation.”

Minor Comments:

2) In the sentence on lines 105-109, the authors note that transcripts with significant differences in TE upon DENR depletion had longer 5'-UTRs than the genome-wide average, which is consistent with the enrichment for uORF containing RNAs in this sample which have longer 5'-UTRs. Is there also any significant difference in the UTR lengths between DENR-regulated and non-DENR regulated uORF containing UTRs? Discussing this could be informative regarding the question why some uORF-containing transcripts are DENR dependent and some are not...

Author response: The Reviewer proposes an interesting analysis that we have now carried out in some detail, see below as **Rebuttal Figure 1**. Indeed, the analysis shows that DENR-responsive transcripts for which we have evidence for translated uORFs (n=156) have longer 5' UTRs than all transcripts with translated uORFs (n=3273). While doing this analysis, we realised that different interpretations of this outcome are possible. First, it could be interpreted by the need for a longer distance between the uORF and CDS start codon in order to convey DENR-dependence. In our previous publication, Castelo-Szekely et al., 2019 (PMID 30982898), we had used a modelling approach to identify predictors of DENR-dependence and had indeed found that a longer uORF-CDS intercistronic distance was significantly associated with MCTS1-DENR responsiveness. Second, in the same publication we had also found that DENR-dependence positively correlated with the number of uORFs found within the 5' UTR – possibly a trivial outcome indicating that if there are more uORFs, it is also more likely to have at least one that fulfils the criteria for MCTS1-DENR-dependent re-initiation. Further analysis on our datasets has confirmed this possibility. Thus, the DENR-responsive transcripts contain more uORFs on average (3.36 uORFs per transcript) than all translated uORF-containing transcripts (2.28 uORFs per transcript). Since the number of uORFs itself is further known to correlate with the length of the 5' UTR, see e.g. Iacono et al., 2005 (PMID 15777708), it is quite possible that the greater UTR length of DENR-responsive transcripts is the reflection of the higher uORF content.

Rebuttal Figure 1 contains the complete analysis which, however, we felt would be too detailed for the actual manuscript, as it would complicate the flow of our story. In Figure 1C of the main manuscript, we thus now show an updated, yet simpler representation, and we describe the outcome in the figure legend rather than in the main text. We hope

that the Reviewer agrees with this decision; if wished, we could certainly also put the more extensive analysis in the main paper.

3) The beginning of the results section discusses translational efficiency on the main ORF as a result of MCTS1-DENR depletion. Is it possible to say from the data whether translational efficiency/ footprint density on the uORFs themselves also changes? This could inform our understanding of what these factors actually do during re-initiation.

Author response: This is an interesting question that we have attempted to address by dedicated analyses. We have taken the ensemble of transcripts containing translated uORFs for this analysis and compared the uORF translation efficiency for every translated uORF individually (i.e., normalised RPF counts mapping on the uORF, relative to RNA counts from the whole transcript) in *Denr*-deficient vs. control cells. We then plotted as two separate groups the outcome for uORFs on *Denr*-responsive transcripts vs. those on all transcripts. The resulting distributions are shown in **Rebuttal Figure 2A**. In addition we did a second analysis; here, instead of individually quantifying every uORF translation efficiency (which suffers of low coverage and noisy data in many cases, given that uORFs are typically very short and have few footprint reads), we quantified all 5' UTR-mapping reads, as a proxy for aggregate uORF translation (as mentioned above, many transcripts that we analyse actually contain >1

uORF). The latter analysis is shown in **Rebuttal Figure 2B**. Both analyses show similar outcomes. Thus, we detect globally slightly fewer uORF-mapping footprints when DENR is absent (i.e., the medians of the distributions are < 0), and this effect is seen in particular for the DENR-responsive transcript group (median of green distribution is further shifted to the negative than median of pink distribution).

We would consider the observed effects not strikingly strong. Moreover, there could be several possible biological explanations. First, it is possible that this result reflects reduced re-initiation efficiency among/between individual uORFs of transcripts containing multiple uORFs. Second, it could also be possible that in the absence of DENR, there is a recycling defect at uORF stop codons that leads to longer dwelling of 40S subunits at the termination codon, which would sterically impede further initiation at the uORF start codon, thus lowering the translation of the uORFs. Please also note, however, that we may not be reliably and quantitatively capturing the footprints from the stop codon-dwelling 80S ribosomes themselves, as they are not stabilised by the elongation inhibitor cycloheximide that we add to our extracts during cell lysis and footprint preparation. Probably other hypotheses and scenarios can be envisaged as well with regard to the biological meaning of the lowered uORF translation efficiency. Thus, given that these observations are not very striking and lack a clear and conclusive interpretation, we think it would be premature and counterproductive for the main thrust of our story to include them in the manuscript.

As this rebuttal document will be publicly available side-by-side with our manuscript, the analysis will still be accessible to interested readers who wish to dig more deeply into our analyses. We hope that the Reviewer agrees with this solution.

4) In figure 2B, the supernatant fraction (2) seems enriched for eIF2D. Do the authors know what causes this effect? I wondered whether this is related somehow to the fact that the general control pathway has been activated (as the eIF2 phosphorylation evidences), but struggle to come up with a mechanism that could increase protein amounts on the relatively short timescales of the preparation of the cell lysate.

Author response: The observation made by the Reviewer is correct, i.e. we see that eIF2D is better recovered in the lysate preparation than other components of the translation machinery that we tested by immunoblot, such as the ribosomal protein

RPL23 that we show in Figure 2B, as well as MCTS1 and DENR. We do not think that the activation of control pathways is relevant here, because increased eIF2α phosphorylation is not yet occurring at this step, but only later, when the *in vitro* translation reaction is carried out. To make this point clearer, we have added the eIF2α-phosphoSer51 western blot to Figure 2B, as also shown here in **Rebuttal Figure 3**.

We do not have any evidence regarding the mechanism that allows eIF2D to be more readily found in the supernatant, and MCTS1-DENR more so in the pellet. We would like to point out that our extract preparation

method is deliberately mild, i.e. we use no detergents and after spin-down we tolerate a large pellet that contains many non-lysed cells, organelles, membranes, etc. (as seen in lane 3 in the figure). One could speculate that ribosomes engaged in membrane-associated translation (at the endoplasmic reticulum) are less well recovered in our protocol, and that relatively more MCTS1-DENR than eIF2D is associated with such translation events, hence giving rise to differences in levels in the supernatant. One could also speculate that eIF2D is generally less associated with other macromolecules, proteins, and structures under steady-state conditions in our cells and therefore easily goes into the soluble fraction, whereas MCTS1-DENR is more quantitatively associated with structures that gives it a disadvantage in terms of solubility and recovery. In the future, we hope to be able to unravel the actual activity and determine the interaction partners of eIF2D, which would allow us to put together mechanistic models that will then also be able to explain the observation made by the Reviewer.

5) In figure 2D, firefly luciferase activity in the replicates seems highly variable (over more than one order of magnitude, much more so than the Renilla luciferase used for normalisation). Why is this - did these samples come from different conditions? Within-sample variability in all subsequent analyses seems much less variable.

Author response: In this figure, we plotted raw bioluminescence signals from an array of experiments that we carried out over several months. The variability in firefly luciferase signals is associated with different *in vitro* translation lysate batches that we prepared and used over time – some preparations that we made showed generally higher activity than others. It is indeed interesting that the signal for the Renilla reporter is much less affected by the lysate batch. One possible explanation could be that the Renilla luciferase reporter mRNA is “simpler” and possibly more robust; in particular, it has a very short 5' UTR of only 11 nt, thus probably harbouring fewer potential regulatory motifs. Importantly, across all experiments in the manuscript, we always internally normalise with the control reporters (“no uORF” = 100%) using the same lysate batch, removing this source of variability. This explains why in subsequent analyses the data gives consistent outcomes.

6) On line 152, it would be useful to briefly mention what GADD34 actually is.

Author response: According to the Reviewer’s suggestion, this sentence now reads in lines 168-170:

“As previously reported [25, 26], the addition of recombinant GADD34 Δ 1-240 (a protein that recruits protein phosphatase 1 to eIF2 α to induce its dephosphorylation) greatly counteracted Ser51 phosphorylation (**Figure 2C**).“

7) On line 64, the authors mention conservation of MCTS1-DENR in "Drosophila, mouse and human". On line 66, yeast orthologues are mentioned - it would slightly improve reading flow if yeast was also listed in the first sentence.

Author response: The context of the original sentence was as follows:

“Heterodimeric MCTS1-DENR is a non-canonical initiation factor conserved from yeast to mammals and has been identified as the first, re-initiation-specific factor with conserved activity in *Drosophila*, mouse and human [10, 11, 15-18]. *In vitro*, MCTS1-DENR shows the double activity of being able to recruit and release tRNA from the ribosomal P-site [14, 18]. The yeast orthologues, Tma20-Tma22, appear to mainly function in post-termination tRNA removal and 40S ribosome recycling rather than as promoters of re-initiation [19, 20]. In animal model systems, DENR loss-of-function followed by ribosome profiling has been used to identify CDS translation efficiency changes indicative of altered re-initiation, thus identifying likely direct MCTS1-DENR targets [10, 11, 17, 21].“

Given that there is little evidence that Tma20-22 promotes re-initiation in yeast (and, instead, rather seems to inhibit re-initiation in some cases – see preprint Jendruchová et al. 2024; PMID 38903097) – we still believe that it is correct in the context of the whole paragraph to mention conservation of re-initiation activity specifically for the

metazoan species. Still, the proteins themselves are also conserved in yeast. In order to help the flow of the text, we have slightly changed the paragraph, allowing us to accommodate the Reviewer's suggestion of mentioning yeast upfront. Now, it reads as follows (lines 69-79):

"Heterodimeric MCTS1-DENR is a non-canonical initiation factor conserved from yeast to mammals and has been identified as the first, re-initiation-specific factor with conserved activity in *Drosophila*, mouse and human [10, 11, 15-18]. *In vitro*, MCTS1-DENR shows the double activity of being able to recruit and release tRNA from the ribosomal P-site [15, 19]. These findings have led to the idea that MCTS1-DENR may either act to remove the remaining, deacylated tRNA from the P-site post-termination and after subunit splitting, thereby allowing the 40S subunit to regain the scanning competence that is a prerequisite for re-initiation; and/or to mediate the subsequent delivery of a new Met-loaded initiator tRNA molecule to the 40S subunit. The yeast orthologues, Tma20-Tma22, appear to mainly function in post-termination tRNA removal and 40S ribosome recycling without promoting subsequent re-initiation [20, 21]. In animal model systems, DENR loss-of-function followed by ribosome profiling has been used to identify CDS translation efficiency changes indicative of altered re-initiation rates, thus identifying likely direct MCTS1-DENR targets [10, 11, 18, 22]."

Significance (Required):

In my view this is an interesting study that enhances our understanding of re-initiation on upstream ORFs, ubiquitous and important regulatory elements in eukaryotic mRNAs. The addition of *in vitro* translation-based assays to the available toolset (which is not commonly used to study uORF re-initiation) is a substantial contribution, and in this manuscript is combined with omics-based *in vivo* analyses to advance our understanding of specific re-initiation factors. The confirmation that MCTS1-DENR is a specific re-initiation factor on a subset of human RNAs, that eIF2D can support re-initiation but appears to have roles outside of this process *in vivo*, and that MCTS2 is a functional paralogue of MCTS1 are novel findings of interest to a wide scientific community.

Author response: We wish to thank the Reviewer for the insightful and constructive evaluation of our manuscript. We hope we have been able to address the remaining issues in a satisfactory fashion.

Reviewer #2

Evidence, reproducibility, and clarity (Required):

Summary:

Translational reinitiation, a process of ribosome restart scanning of mRNA segments after termination of short ORF, has been broadly recognized in many transcripts. In addition to regular translation initiation factors, protein players specifically contributing to the reinitiation have been proposed: MCTS1-DENR complex and eIF2D. However, the quantitative assessment of these factors to reinitiation remained elusive.

The authors addressed MCTS1-DENR-mediated translational reinitiation by an elegant *in vitro* translation system in humans. This system faithfully recapitulated the uORF-dependent impacts of these factors on cellular translation, observed by ribosome profiling. Importantly, the authors quantitatively dissect the ribosome flux for leaky scanning to bypass uORF, reinitiation from uORF, and ribosome recycling after uORF translation. Strikingly, MCTS1-DENR contributes to the reinitiation process in a significant fraction, whereas eIF2D has marginal effects on that. The author also found that MCTS2, a paralog of MCTS1, has a similar function to MCTS1 for reinitiation.

Major comments:

This manuscript is well written. The data strongly support the conclusion. This reviewer does not have a major comment for this manuscript. This reviewer has only several minor comments as described below.

Author response: This is an exceptionally positive evaluation and we are very grateful to the Reviewer for the comments that we have all addressed below and that have allowed us to further improve the manuscript.

Minor comments:

1. In line 102, "non-functional (nf) Eif2d shRNA" sounds a bit confusing. Although this reviewer guessed that "non-functional" shRNA may be designed by shuffled sequence or so, the another wording may be easy for readers.

Author response: We have now simplified the nomenclature and have changed throughout the text, figures and figure legends, to con1 and con2 shRNA for the two different control shRNAs. Information on the identity of the two control shRNAs has been moved to the Materials and Methods section.

2. For Figure 1C, the caption for "Denr shRNA" looks a bit confusing. It should be "transcripts with low TE by Denr shRNA" or so.

Author response: We agree with the Reviewer's comment. We have now changed the caption to "*Denr*-responsive transcripts (low TE)".

3. In Supplementary Figure S3A, the authors treated cell lysate with tunicamycin and recapitulated UPR/ISR mediated by PERK on ER. This sounds interesting since even in the lysate, ER may be intact for UPR/ISR. The author should consider commenting on this in the text.

Author response: We believe that there has been a misunderstanding and we would like to apologise if we did not explain the experiment and label the figure sufficiently clearly in the previous version of the manuscript. Thus, in this experiment, HeLa cells were treated with tunicamycin for 14h before harvesting cells/lysate for the analyses. We did not treat translation-competent lysate post-harvest with tunicamycin and, to be honest, we would not expect to be able to specifically activate an ER- and translation-dependent stress response pathway in the *in vitro* setting. Probably something to

explore and investigate in the future, but that said, this is not an easy experiment to analyse in our system because we know that *in vitro* translation in the absence of GADD34 Δ 1-240 activates eIF2 α phosphorylation already *per se* (see Figure 2B-C). Hence, it may be difficult to disentangle effects from tunicamycin from those that occur anyway in the extract.

To avoid misunderstanding of this experiment by future readers, we have introduced an additional mention of the “*in vivo*” character of the treatment in the text as follows, and we have also changed some of the labelling of this Figure (which has now become Supplementary Figure S4) to make things clearer. See lines 253-255:

“We thus carried out our assays in the presence vs. absence of GADD34 Δ 1-240, which led to low and high Ser51 phosphorylation levels of eIF2 α , respectively (**Figure 2C**). Of note, the level of Ser51 phosphorylation occurring in the absence of GADD34 Δ 1-240 *in vitro* was even higher than that seen *in vivo* after a standard ISR induction protocol by tunicamycin treatment of cells (**Supplementary Figure S4A**).”

4. For Figure 3 and Supplementary Figure S3, the authors used the same jargon of “reinitiation” for the effects from both MCTS1-DENR and phosphorylated eIF2 α . However, the points to which these factors contributed, should be different; MCTS1-DENR may function the departure of 40S from termination codons (or blocking 60S rejoining), whereas phosphorylated eIF2 α blocks the recruitment of initiator tRNA on 40S. More concrete wording in the paragraph may be helpful for the understanding of readers.

Author response: Thank you for this comment. Indeed, we were using the word “re-initiation” quite broadly and possibly too indiscriminately for the whole process that would lead to CDS translation after uORF translation, i.e. all the way from the moment when the terminating ribosome commits to engage into partial recycling (= removal of 60S and deacylated P-site tRNA, but retention of 40S) to re-scanning by the 40S, and re-assembly of an initiating 80S on the downstream AUG. We have now added more concrete molecular details to the description, and we have extensively reworked this whole part of the manuscript, which can be found from line 229.

Along similar lines, Reviewer 3 also felt strongly that we should, whenever possible, use more precise terminology to describe the activities and steps that are relevant along the re-initiation process, rather than putting everything under one re-initiation umbrella. Throughout the manuscript we have now tried to implement a better description. These changes are all track-changed in the manuscript file.

5. Related to the point above, the author may consider explaining why Atf4 and Atf5 are more sensitive to phosphorylated eIF2 α than Asb8 and Klhdc8a.

Author response: This is a very interesting question we wish to address in future work. At the moment, we can only speculate on why *Atf4* and *Atf5* are so clearly dependent on the activity of the ternary complex, yet the effects on *Klhdc8a* and *Asb8* are modest in our experiments. Are *Atf4* and *Atf5* the “odd ones out”, and contain a specific configuration (i.e. the two uORFs and competing re-initiation events?) that make them more eIF2 α -dependent? Or are *Klhdc8a* and *Asb8* exceptions? In the long term, we

are planning to devise other, additional and more specific reporters to address this question – but this would be the topic of a new, dedicated study, rather than fall within the scope of the current manuscript. As we cannot offer the reader a very good explanation for the intriguing differences in the context of our manuscript, we refrain from too much speculation. Instead, we end this part of the results section with the following summary and outlook, see lines 269-274:

“[...] Taken together, these results suggest that eIF2 α activity is only moderately limiting for re-initiation *per se*, as judged from our ‘single uORF’ reporters, *Klhdc8a* and *Asb8*. One may speculate that it is in particular in the context of ‘multiple uORF’ configurations and competing re-initiation events – i.e., in cases such as on *Atf4* and *Atf5* – that the influence of phospho-eIF2 α comes to bear. Our results can serve as a starting point for future investigations into possible differential TC requirements across different uORF-dependent re-initiation events.”

6. For Figure 5H, sole DENR could partially rescue reinitiation deficiency in DENR KO lysate which also lost MCTS1. Although this reviewer guessed that this originated from the residual amount of MCTS1 in the lysate, the author should consider adding an explanation for this.

Author response: Indeed, recombinant DENR alone showed partial rescue, and recombinant MCTS1-DENR or MCTS2-DENR full rescue. In the DENR KO, MCTS1/2 levels are greatly reduced due to co-dependence for protein stability, yet we do still see very low levels remaining, see immunoblot in Figure 1H (formerly 1J). Therefore, it is indeed our hypothesis that the ability of recombinant DENR alone to partially rescue relies on its association with remaining MCTS1 or MCTS2 protein. We have added this information in lines 396-400 which now read:

“To gain insights into whether MCTS1-DENR and MCTS2-DENR could redundantly act in re-initiation, we tested both as recombinant proteins on *Klhdc8a* and *Asb8* reporters. Addition of recombinant DENR alone showed only low rescue activity, likely due to residual MCTS1/2 in *Denr* KOs (see **Figure 1H**), yet supplementing with either of the MCTS paralogs together with DENR led to a comparable and full rescue (**Figure 5H, I**).”

7. The author should add a Western blot for MCTS2 in DENR KO cells to test whether DENR loss also affects the stability of MCTS2 as seen in MCTS1.

Author response: Thank you for this suggestion, which has allowed us to obtain new interesting insights. To address this point, we would need MCTS2-specific antibodies, which are currently not available from a commercial or other source. However, because MCTS1 and MCTS2 only differ at a few amino acid positions, it is actually possible that the antibody we are using to detect MCTS1 – raised against the full-length recombinant MCTS1 protein (Ahmed et al., 2018; PMID 29889857) – recognises both proteins, MCTS1 and MCTS2. To test if this was the case, we used immunoblots against the MCTS1 and MCTS2 recombinant proteins that we supplemented into *Denr/Eif2d* double KO *in vitro* translation lysates at the concentrations used in our assays. The result of this experiment is shown in **Rebuttal Figure 4**. The experiment shows very clearly that the anti-MCTS1 antibody is able to recognise MCTS1 and MCTS2 with similar efficiency. We deem it likely that some of the other, commercially available anti-MCTS1 antibodies also recognise both paralogs

– meaning that in reality, previous publications may have already reported on the ensemble of MCTS protein levels rather than on MCTS1 alone.

To come back to the Reviewer's point "[...] whether DENR loss also affects the stability of MCTS2 as seen in MCTS1.", we can thus respond positively to the question: given that overall MCTS signal is greatly reduced in *Denr* KO (Figure 1H), both MCTS paralogs suffer the same fate and become destabilised.

We believe that these findings could be of interest to the reader and research community and we have therefore added this western blot as Supplementary Figure S6B and refer to it accordingly, right at the end of the results section.

Significance (Required):

Translational reinitiation, a process of ribosome restart scanning of mRNA segments after termination of short ORF, has been broadly recognized in many transcripts. In addition to regular translation initiation factors, protein players specifically contributing to the reinitiation have been proposed: MCTS1-DENR complex and eIF2D. However, the quantitative assessment of these factors to reinitiation remained elusive.

The authors addressed MCTS1-DENR-mediated translational reinitiation by an elegant *in vitro* translation system in humans. This system faithfully recapitulated the uORF-dependent impacts of these factors on cellular translation, observed by ribosome profiling. Importantly, the authors quantitatively dissect the ribosome flux for leaky scanning to bypass uORF, reinitiation from uORF, and ribosome recycling after uORF translation. Strikingly, MCTS1-DENR contributes to the reinitiation process in a significant fraction, whereas eIF2D has marginal effects on that. The author also found that MCTS2, a paralog of MCTS1, has a similar function to MCTS1 for reinitiation.

This work provides important findings for broad readers not only in the RNA/translation field but also in immunity and neurological disorders in which MCTS1-DENR is involved.

Author response: It has been a pleasure to address the Reviewer's insightful comments and we are grateful for the inputs that have allowed us to significantly improve our manuscript.

Reviewer #3

Evidence, reproducibility and clarity (Required):

Meurs et al. provide evidence for the role of DENR and MCTS1, as well as a the MCTS2 paralog, in translational control regulated by re-initiation after uORFs. These studies are based off of reporter assays in mammalian in vitro translation lysates prepared from WT or depleted cells, as well as ribosome profiling and RNA-seq. The experiments are well controlled, appropriate for the questions being asked, and incorporate rigorous controls and methods. The manuscript does provide new insight into DENR, MCTS1, MCTS2, and eIF2D that is informative for the field. However, my interpretation of the tone and model from the authors is that DENR and MCT1/2 are directly acting in re-initiation (i.e., at the re-initiated start codon)- however, their data can also be explained by the counter model in the field that these factors are in fact ribosome recycling factors. This does not lower the validity of these new data, but I feel they should be presented with both models in mind and equally plausible. From what I can tell, the authors have not tested the two models; I don't think they need to at the moment but they should present both models equally.

Author response: We appreciate the Reviewer's insightful, balanced and positive comments. The various points of criticism are all well-taken and we have addressed them as outlined below. In particular, we fully understand that in terms of terminology, we were a little bit too simplistic, using the term "re-initiation" indiscriminately for the whole process that would eventually lead to the actual new initiation event on the downstream start codon, i.e. encompassing recycling, scanning, and initiation at the downstream start. This can indeed be misleading (e.g. see also Reviewer 2, point 4) and we have therefore revised the whole manuscript accordingly.

My comments are:

1. The authors reference published structural work supporting that these alternative initiation factors in fact do directly act in re-initiation (at the re-initiated start codon); however, these studies used a truncated/mutated viral IRES and the overall conclusion has not been shown directly on any mammalian mRNA. I apologize if I missed the latter, but if so, please directly state this and the evidence.

Author response: The structures available from the literature are reconstituted from purified components – 40S, MCTS1-DENR, initiator-tRNA, viral IRES construct. Therefore, they indeed represent an artificial situation. In particular the viral IRES is artificial, as pointed out by the Reviewer, but also the initiator-tRNA: probably other

tRNAs would equally assemble into analogous structures had they been used in the reconstitution instead of initiator-tRNA.

We have now made the Reviewer's point regarding the IRES RNA clearer at several instances in the manuscript, e.g. in the Results section lines 386-391:

"Using available structural data from *in vitro* reconstituted MCTS1-DENR in complex with the 40S ribosome, an initiator-tRNA molecule and a viral IRES-containing model RNA [35] (please note that no such structure has so far been reported for a natural, non-viral transcript), we found that MCTS2-specific amino acid changes affected positions that all cluster in the same region of the fold but outside of the interaction surface between MCTS1-DENR and also outside of that between MCTS1-DENR and the 40S ribosomal subunit (**Figure 5G**)."

And, similarly, in the Discussion section from line 445:

"This evidence was, mainly, [...], (ii) the *in vitro* ability shared by both factors to act on 40S subunits and deliver Met-tRNA^{Met}_i to the empty P-site ([15]; using an experimental model where 40S is in complex with HCV IRES-like mRNA – it should be noted that proof of equivalent structures on cellular mRNAs or *in vivo* is still lacking); [...]"

2. Figure 1 layout is not too intuitive as it goes in different directions; additionally, 1J is referenced in the text before 1H and 1I.

Author response: We have changed the order of panels H, I, J – they are now referenced in the correct order in the text. We have also rearranged the panels, which we believe has rendered the figure layout more intuitive.

3. In Figure 1F, the increase in RPFs in the Denr KD is highly consistent with a defect in ribosome recycling, no re-initiation per se. This appears to be glanced over and not tied into any of the downstream hypotheses. (see below regarding comment about ribosome recycling from PMID 30146315 and 34016977).

Author response: We have now added a sentence to make this clearer, see lines 127-130:

"Visual inspection of footprint distributions on the transcripts, indicating decreased coverage on the CDS and comparable or even increased coverage on the uORF (**Figure 1G**), were compatible with MCTS1-DENR acting in ribosome recycling and/or re-initiation itself."

4. Just to avoid potential issues as some Denr, MCTS1 and eIF2D studies as done in animal models, the use of "in vivo" in this paper should be more accurately described as "in 3T3/HeLa cells"

Author response: In the revised version of the manuscript, we have now generally added this more accurate information on cells where we formerly had written "*in vivo*".

5. In Figure 2D, the use of two subtle gray tones that overlap makes the data incomprehensible. I assume the data on the left in each major group (RLuc and FFLuc) is +GADD34?

Author response: We apologise for the poor colour choice. We have changed the colours to accommodate the Reviewer's criticism (+GADD34 is now shown in grey, and -GADD34 in yellow in Figure 2D).

6. For line 239-240, the line "Taken together, these results suggest that eIF2 α levels are only moderately limiting for re-initiation." should be more accurate. For example, "Taken together, these results suggest that phosphorylated eIF2 α levels are only moderately limiting for re-initiation on *Klhdc8a* and *Asb8* mRNAs in vitro."

Author response: We have now changed this sentence and added a bit more perspective in response to a related comment from Reviewer 2 (point 5). Now this part reads (lines 269-274):

"[...] Taken together, these results suggest that eIF2 α activity is only moderately limiting for re-initiation *per se*, as judged from our 'single uORF' reporters, *Klhdc8a* and *Asb8*. One may speculate that it is in particular in the context of 'multiple uORF' configurations and competing re-initiation events – i.e., in cases such as on *Atf4* and *Atf5* – that the influence of phospho-eIF2 α comes to bear. Our results can serve as a starting point for future investigations into possible differential TC requirements across different uORF-dependent re-initiation events."

7. The line on page 10, "This measure is expected to increase when a re-initiation factor is depleted because ribosomes are selectively lost from CDS sequences but will still translate uORFs when re-initiation is inhibited [17]." Isn't this more consistent with role of MCTS1 and DENR in 40S recycling at stop codons (not re-initiation *per se*) as shown by the Guydosh Lab? They have also shown that MCTS1 and DENR play a bigger role than eIF2D in this role, at least in yeast-PMID 30146315 and 34016977. In general, I feel this point should be brought up in the start of the manuscript. While I don't disagree with the included data, but the impact on recycling at the identified uORFs (not direct re-initiation at the main/primary ORF start codon) can also easily explain the observed effects with DENR and MCTS1.

Author response: We have now carefully overhauled the manuscript to rework the phrasing that we use and to explain the different mechanistic possibilities, i.e. the role in recycling/tRNA removal as opposed to a direct role in the re-initiation event itself. All the modifications in this regard can be seen in the track-changed manuscript.

Moreover, we have now also specifically taken a look at ribosome occupancy at termination codons and generated metagene plots similar to those shown by the Guydosh lab on the yeast recycling phenotype. These are shown in **Rebuttal Figure 5**. Panel A is an excerpt from the Young et al. 2021 publication, showing that the recycling defect leads to an accumulation of queued ribosomes and low stop codon occupancy in the yeast 80S footprint data. An analogous analysis on our *Denr* shRNA and our *Eif2d* shRNA data, shown in panel B, indicates that this phenotype is not prevalent in our mammalian dataset. While we see higher stop codon signal in the absence of DENR, there is no indication of queuing. An analysis of the subset of *Denr*-responsive-genes does not reveal any queuing either (panel C). Specific analysis at

uORF stop codons transcriptome-wide (panel D) or on *Denr*-responsive transcripts (panel E) again shows the higher stop codon signal (indicated by the * symbol).

We decided not to put these analyses into the actual manuscript because we feared it would become a collection of too many individual, "all over the place" pieces of data and distract from the main thrust of our story. However, if the Reviewer wishes, we can certainly still integrate it in the Supplementary Figures. We would like to point out that this rebuttal document will also be publicly available on Review Commons, and the analyses are thus available for the interested/specialised readership.

8. The authors should provide evidence that their recombinant proteins used in this study are highly pure.

Author response: We provide evidence below, in **Rebuttal Figure 6**. We have also added the panels to the manuscript as Supplementary Figure S2D, E.

9. Cricket Paralysis Virus IRES should be Cricket Paralysis Virus IGR IRES. CrPV has two reported IRESs. The IGR is the "factorless" one.

Author response: Thank you for the correction. We have implemented it.

10. In the Discussion, the critique of the eIF2D and related data in the field has been recently described elsewhere too-PMID 38433101.

Author response: We would like to thank the Reviewer for the pertinent comment and have now included this reference to our manuscript.

Significance (Required):

The manuscript does provide new insight into DENR, MCTS1, MCTS2, and eIF2D that is informative for the field. However, my interpretation of the tone and model from the authors is that DENR and MCT1/2 are directly acting in re-initiation (i.e., at the re-initiated start codon)- however, their data can also be explained by the counter model in the field that these factors are in fact ribosome recycling factors. This does not lower the validity of these new data, but I feel they should be presented with both models in mind and equally plausible. From what I can tell, the authors have not tested the two models; I don't think they need to at the moment but they should present both models equally.

Author response: We have systematically gone through the manuscript to try to eliminate all instances of bias. As mentioned before in this rebuttal document, we were using the term “re-initiation” for the whole *process* allowing CDS translation post-uORF-termination. We have now used more specific terminology throughout, and we hope that the new version finds the Reviewer’s approval.

Reviewer #4

Evidence, reproducibility and clarity (Required):

Summary: In this manuscript, Meurs and colleagues describe a multifaceted analysis of the roles of eIF2D, DENR, MCTS1 and MCTS2 in translation reinitiation. Translation reinitiation occurs primarily after ribosomes translate short upstream Open Reading Frames (uORFs). Previous work has established that DENR/MCTS1 contributes to reinitiation after translation of many uORFs, to varying extents. It has been suggested that eIF2D is also a reinitiation factor due to structural similarity to the MCTS1/DENR complex. Meurs et al used a clever combination of in vivo and in vitro reporter assays and ribosome profiling to evaluate the roles of these factors on reinitiation after translation of several uORFs. The manuscript is well written, addresses an important knowledge gap in translation, and uses a satisfyingly elegant set of experiments to quantitate leaky scanning and translation reinitiation. The main novel findings are that the reinitiation function of DENR/MCTS1 varies across uORFs, that eIF2D can partially substitute for DENR reinitiation activity in vitro, and that MCTS2 is expressed and functions in reinitiation. MCTS2 is an intronless gene that was created by retrotransposition, and was previously reported to be a non-functional pseudogene. Overall, there is much to appreciate, with only a few areas that could be addressed.

Author response: Thank you for this positive and encouraging evaluation.

Numbered comments follow:

1) Figure 2J - The authors say that lengthening the ASB8 uORF appears to reduce DENR dependence for reinitiation. Is this difference statistically significant?

Author response: We have now carried out 2-way-ANOVA to test if DENR-dependence was specifically changed by lengthening the uORF by TGT insertion. We find significant interaction ($p = 0.021249$) between genotype and uORF lengthening, indicating that the difference between wild-type and *Denr* KO lysates for the 1-aa uORF is significantly different from the one of the 2-aa TGT uORF. We have added this information to the figure legend.

For the record, we do not see a significant interaction in the other comparison, i.e. 1-aa uORF vs. the 2-aa uORF carrying GCG insertion; here, genotype explains all difference.

2) As the authors acknowledge, the lack of apparent translation efficiency changes in the eIF2D KD / deletion may result from its activity being redundant with DENR-MCTS1. Do mRNA levels or translation of MCTS1, MCTS2, or DENR change with eIF2D deletion or depletion?

Author response: We have done this analysis and show it below in **Rebuttal Figure 7**. Of note, we do not see any upregulation of *Mcts1*, *Mcts2* or *Denr* in *Eif2d* shRNA-treated cells that could point to a compensation mechanism. Also, our analyses of protein levels by immunoblot (main Figures 1A and 1H) do not suggest that these factors are massively changed.

3) It's a little disappointing that the authors can't clearly determine whether eIF2D functions in reinitiation, in part because eIF2D deletion or depletion lead to a cell cycle defect. They suggest future work with an auxin degree version of eIF2D and depletion in the cell. Could they also use immunodepletion in extracts to more directly evaluate eIF2D in this study?

Author response: Thank you. We are also very disappointed that after years of experiments and trying many approaches, we still do not have a positive answer as to what eIF2D is truly doing in cells! We have considered immunodepletion previously; however, first immunoprecipitation tests with the available antibodies indicated that IP of the native protein does not work – probably the epitope is not accessible. Therefore, immunodepletion would be difficult to achieve. We have now invested a lot into establishing degron-tagging technology which will hopefully allow us to come to a breakthrough in future experiments.

4) The graphed reporter results include shaded areas representing uORF repression, leaky scanning, reinitiation, and DENR dependence. While this is very concisely presented, it is also difficult to read the graphs. Could the authors provide a mini bar graph (perhaps log scale) to show this more clearly?

Author response: We have added such bar graphs (after trying several display options, we decided on linear scale) for all the *in vitro* experiments from the main Figures. They can be found in Supplementary Figure S3.

5) On line 233, the authors describe the Klhdc8a uORF-CDS distance as 73 nucleotides, but it looked shorter to this reviewer (60 nt). Could you please double check that?

Author response: We doubled-checked the sequences and can confirm that the distance is 73 nt. In Supplementary Figure S1 we document the sequence used in the reporter and also show an alignment across different mammalian orthologs. Indeed, in several species (including humans) the distance is slightly shorter. Maybe this was the source of the misunderstanding. Our reporters are based on the murine version of the gene.

6) It is great to see the authors checked the eIF2-alpha phosphorylation state in their *in vitro* extract preparations. However, in some ways this opens up a "can of worms" regarding *in vitro* translation systems in general. One can't check all stress-related modifications. For example, are ribosomes ubiquitinated due to the RQC pathway activation in extracts? Are any other initiation factors post-transcriptionally modified? This is not necessarily a criticism of this paper or its use of *in vitro* translation, but it is more a comment on *in vitro* translation systems in general. The authors may (or may not) want to add this to the discussion.

Author response: This is a pertinent question and worth a dedicated analysis in the future – indeed, so many things one could/should check in terms of potentially modified or activated pathways...!
We considered adding a paragraph on this to the discussion but decided against it for now because we do not have additional data, for example, on RQC, and we also do

not want to claim being the first to observe the change in eIF2 α phosphorylation state that happens during *in vitro* translation reactions. The paper that originally suggested adding GADD34 Δ 1-240 to *in vitro* translation reactions to reduce phosphorylation, Mikami et al., 2010 (PMID 20349333), shows exactly this effect in their Figure 2b (see here: <https://link.springer.com/article/10.1007/s10529-010-0251-7/figures/2> : “Cont” is the pure cell extract without undergoing *in vitro* translation, and “no GADD34” is cell extract after *in vitro* translation; very clearly there is the same phosphorylation going on as in our experiments).

Based on the Mikami et al. publication, Gurzeler et al., 2022 (PMID 34965175), adapted the idea of adding GADD34 to the reactions (although to the best of our knowledge they indeed do not show data on eIF2 α phosphorylation increase occurring *during* the reaction). The latter paper, Gurzeler et al., is what triggered our interest in setting up a dedicated *in vitro* system for re-initiation.

Significance (Required):

General Assessment: Overall, this manuscript reports very high quality work. Experimental design was excellent, data analysis was well done, the manuscript is well-written and the figures are very clear for the most part.

Advance: The main advance of the paper is that it shows that eIF2D can replace the function of MCTS1-DENR and that MCTS2 is a functional gene. The *in vitro* assay system is a modest technological advance (*in vitro* translation systems are commonly used), but it is highly appreciated that the authors checked eIF2-alpha phosphorylation in the extract.

Audience: The audience for this paper would be primarily a gene expression / mRNA translation audience.

Author response: We would like to explicitly thank the Reviewer for the constructive and very positive comments.

Dear Dr Gatfield,

Thank you for submitting a revised version of your manuscript and for the productive discussions regarding the concerns raised by the referees during the previous review at review commons. We have shared your point-by-point response with the original referees who find that their previous concerns have been addressed and now recommend publication of the manuscript. There remain only a few mainly editorial points that have to be addressed before I can extend formal acceptance of the manuscript:

- Please include the funding information in the "Acknowledgements" section.
- On the abstract page of the manuscript, please include 4-5 general keyword terms to enhance searchability.
- Please adjust the format of the reference list and of the in-text citations according to EMBO Journal format (alphabetical order, author name et al + year.../up to 10 author names in the reference list before et al / please refer to our Guide to Authors for additional information on EMBO J reference format).
- Please rename the "Data and script availability" section to "Data Availability".
- Please rename the Conflict of Interest section into "Disclosure and Competing Interests Statement", in accordance with our updated Guide to Authors (<https://www.embopress.org/competing-interests>)
- As we are switching from a free-text author contribution statement towards a more formal statement based on Contributor Role Taxonomy (CRediT) terms, please remove the present Author Contribution section and instead specify each author's contribution(s) directly in the Author Information page of our submission system during upload of the final manuscript. See <https://casrai.org/credit/> for more information.
- Please provide either a "Yes" or a "Not Applicable" answer to each one of the questions in your Author Checklist (<https://www.embopress.org/pb-assets/embo-site/EMBO%20Press%20Author%20Checklist-1642513524327.xlsx>). In the last column of this checklist, only the sections of the manuscript where the relevant information can be found should be listed (the information per se should be included in the main manuscript file).
- Please upload the supplementary figures as individual files (Regarding the supplementary figures, you can upload up to 5 as EV Figures, in which case they need to be uploaded as separate Figure files as well, with their legends in the manuscript file, after the main figure legends. The remaining figures should be compiled in one PDF file labeled "Appendix" with their legends. Please ensure that the figure legends are included, and that the appendix has a table of contents with page numbers. Please also ensure that the nomenclature for the figures and tables is correct, i.e. "Appendix Figure S1" and "Appendix Table S1".)
- Please update source file names, titles, legends and manuscript callouts to Dataset EV1-EV2 instead of Supplemental Table S1-S2; legends should be uploaded as a separate tab/sheet in each Excel file
- Please provide the Reagent and Tools Table. For more information, please check <https://www.embopress.org/page/journal/14602075/authorguide#structuredmethods> and download the template for Reagent Table (https://www.embopress.org/pb%2Dassets/embo-site/Reagents_Tools_Table_TEMPLATE.docx)
- Please provide suggestions for a short 'blurb' text prefacing and summing up the conceptual aspect of the study in two sentences (max. 250 characters), followed by 3-5 one-sentence 'bullet points' with brief factual statements of key results of the paper; they will form the basis of an editor-written 'Synopsis' accompanying the online version of the article. Please also provide an altered synopsis image, making sure that the aspect ratio conforms to our website's format - it should be exactly 550 pixels wide and between 300-600 pixels high.
- Please include the specific URLs for GSE263991 and PXD051482 datasets in the data availability statement.
- Please provide the exact p values in the legends of figures 1B, C, D, J, 2E, G, I, J; 3A, C; 4B, C; E, H, I, J, K; 5H.
- Please indicate the statistical test used for data analysis in the legends of figures 2I.
- Please note that the p value is not represented in the figure 5E, however statistical test related information is provided in the legend of the corresponding figure.
- Please note that the box plots need to be defined in terms of minima, maxima, centre, bounds of box and whiskers, and percentile in the legends of figures 1F, J; 2I, J; 3A, C; 4E-K; 5E, H, I.
- Please note that information related to n is missing in the legends of figures 1F, J, 2D, E, G, I, J; 3A, C; 4E-K; 5E, H, I.

- Please adjust the section order which should be: Title page - Abstract & Keywords - Introduction - Results - Discussion - Methods - Data Availability - Acknowledgements - Disclosure and Competing Interests Statement - References - Figure Legends - Table(s) - Expanded View Figure Legends.

- Please make sure to provide Source data and fill out the Source data checklist which will be files sent by my colleague Hannah Sonntag. Please complete the Source Data checklist and upload it to our online system. Source data files need to be saved in a scheme one figure/folder and then uploaded as .zip files. E.g. all the Source data files for figure 1 need to be saved in a single folder and this needs to be zipped and then uploaded as "SD figure 1.zip" file.

With best regards,

Cornelius Schneider

Cornelius Schneider, PhD
Editor | The EMBO Journal
c.schneider@embojournal.org

We realize that it is difficult to revise to a specific deadline. In the interest of protecting the conceptual advance provided by the work, we recommend a revision within 3 months (26th Jan 2025). Please discuss the revision progress ahead of this time with the editor if you require more time to complete the revisions. Use the link below to submit your revision:

Referee #1:

This reviewer confirms that all concerns raised have been addressed and supports the publications in the EMBO Journal.

Referee #2:

The revised manuscript from Meurs et al. has fully addressed my previously raised concerns.

Referee #3:

I originally reviewed this manuscript through Review Commons. I feel the authors have addressed my concerns from the first review, as well as the concerns of other reviewers. I have no additional issues or comments regarding the manuscript.

Referee #4:

The authors have thoroughly and completely addressed all comments I raised during the initial "review commons" round of review. I would like to thank the authors for providing additional experimental evidence in the review where they decided against full inclusion of this evidence in the manuscript, in my view in all cases the decision not to include additional data in the manuscript is perfectly valid.

All editorial and formatting issues were resolved by the authors.

Dear Prof. Gatfield,

I am pleased to inform you that your manuscript has been accepted for publication in the EMBO Journal.

Yours sincerely,

Cornelius Schneider, PhD
Editor
The EMBO Journal
c.schneider@embojournal.org
